# Flash floods versus river floods – a comparison of psychological impacts and implications for precautionary behaviour

Jonas Laudan[1], Gert Zöller[2], Annegret H. Thieken[1]

[1]University of Potsdam, Institute of Environmental Science and Geography, Karl-Liebknecht-Strasse 24-25, 14476 Potsdam, Germany

[2]University of Potsdam, Institute of Mathematics, Karl-Liebknecht-Strasse 24-25, 14476 Potsdam, Germany

*Correspondence to:* Jonas Laudan (jlaudan@uni-potsdam.de)

**Abstract.** River floods are among the most damaging natural hazards that frequently occur in Germany. Flooding causes high economic losses and impacts many residents. In 2016, several Southern German municipalities were hit by flash floods after unexpectedly severe heavy rainfall, while in 2013 widespread river flooding had occurred. This study investigates and compares the psychological impacts of river floods and flash floods and potential consequences for precautionary behaviour. Data was collected using computer-aided telephone interviews that were conducted among flood affected households around 9 months after each damaging event. This study applies Bayesian statistics and negative binomial regressions to test the suitability of psychological indicators to predict the precaution motivation of individuals. The results show that it is not the particular flood type, but rather the severity and local impacts of the event that are crucial for the different, and potentially negative, impacts on mental health. According to the used data, however, predictions of the individual precaution motivation should not be based on the derived psychological indicators, i.e. "coping appraisal", "threat appraisal", "burden", and "evasion", since their explanatory power was generally low and results are, for the most part, non-significant. Only burden reveals a significant positive relation to planned precaution regarding weak flash floods. In contrast to weak flash floods and river floods, the perceived threat of strong flash floods is significantly lower although feelings of burden and lower coping appraisals are more pronounced. Further research is needed to better include psychological assessment procedures and to focus on alternative data sources regarding floods and the connected precaution motivation of affected residents.

# 1 Introduction

In June 2013, i.e. eleven years after the severe 2002 flood event in Germany which caused an overall loss of EUR 11.6 billion (Thieken et al., 2006a), the country was challenged again by strong river flooding, affecting 12 of the 16 federal states, in particular Saxony, Saxony-Anhalt, and Bavaria. Considering country-wide gauge data and peak discharges, the 2013 flood

event can be described as even more severe in hydrological terms than the 2002 flood, yet causing a lower monetary losses of EUR 6 to 8 billion (Thieken et al., 2016a). Additionally, in May and June 2016, heavy rainfall in Central Europe lead to severe surface water runoff, pluvial flooding, and flash floods in Southern Germany. Municipalities in Bavaria and Baden-Wurttemberg were especially affected, resulting in overall losses of EUR 2.6 billion (Munich Re, 2017).

The flash flood events in 2016 were remarkably different from the river flood events of 2002 and 2013 in terms of processes,

dynamics, duration and the type of induced damage on buildings (Laudan et al., 2017). River floods usually occur after long-lasting rainfall or snowmelt within large catchment areas and result in slowly rising water levels. On the other hand, flash floods emerge within (small) catchments where slopes are steep and defined. This leads to rapid, unpredictable flow dynamics that can be rough in terms of a high sediment transport, high flow velocities and forceful discharge. Furthermore, very short lead times between the generating convective system and the resulting flood complicate the forcast of such events and often

do not allow for a timely warning (Borga et al., 2014). Apart from potentially high damage on buildings and infrastructure, flash floods can also cause serious injuries and fatalities (Gaume et al., 2009). Therefore, the occurrence of the severe flash floods across Germany in 2016, outside alpine regions, can be described as unexpected. Still, they are important events to study when the comparatively high monetary losses of EUR 2.6 billion (Munich Re, 2017), and eleven fatalities (four in Baden-Wurttemberg and seven in Bavaria along Simbach am Inn) are considered.

Flood risk management in Germany has a long history with several regulations and ongoing programs, e.g. the "Nationales Hochwasserschutzprogramm" (NHWSP) that was launched after the 2013-flood and national legislation regulation, e.g. the Federal Water Act (WHG). The WHG was first introduced in 1957, revised in 2005, renewed in 2009 and again revised in 2018 to consider gaps that became obvious after damaging flood events. Additionally, regulations such as the European Floods Directive (2007/60/EC) required additional frameworks to be incorporated into the national legislation by 2010 (e.g. Thieken

et al., 2016b). However, despite this, the management of water bodies and flood management are in principle regulated at the state level rather than the national or federal level. After the severe river flood events in 2002 and 2013, flood risk management in Germany and the relevant legislation was carefully revised. In addition to structural precaution measures on a higher level such as dikes and retention areas and non-structural measures such as early warning systems, the focus was shifted to a more integrated flood management approach, in which potentially affected residents are encouraged to prepare themselves. In this

context, private structural precaution measures (i.e. waterproof sealed cellars i.e. dry-proofing, wet-proofing, relocation of heating and electrical utilities) as well as private non-structural flood protection measures (i.e. adapted interior fitting and flood-adapted use such as avoiding water-sensitive furniture in the cellar) became increasingly important (Kienzler et al., 2015; Thieken et al., 2016b; Laudan et al., 2017). Accordingly, the 2005 revision of the WHG designated private precaution as

mandatory which thus requires residents in flood prone areas to undertake appropriate private precautionary actions. As an overall result, regions which have been affected by recurrent river floods are now better managed, having tailored flood risk management plans in place, including private precaution. Still, despite the devastating events in 2016, flash floods and strong surface water runoff are not yet considered as significant national risks and are therefore not treated systematically in the frame

of the European Floods Directive. As a result, little is known about private precaution measures in Germany concerning flash floods and also pluvial floods, which occur in urban areas due to blocked sewer systems after heavy rainfalls. Because of a strong focus on fluvial flooding in recent years, this lack of knowledge remains.

It has been shown that private precaution measures can significantly reduce the mean damage ratio (i.e. the financial flood damage in relation to the total building or content asset value) to households and household contents by up to 53 per cent and

thus play an important role in comprehensive flood management strategies (Kreibich et al., 2005; Thieken et al., 2008; Merz et al., 2010; Hudson et al., 2014). Yet, 15% (in 2002) up to 65 per cent (in 2011) of residents affected by river floods used or furnished their home in a flood-adapted manner (Kienzler et al. 2015; Thieken et al. 2016b). This percentage is lower for residents affected by pluvial floods, i.e. just 7 per cent (in 2014; Spekkers et al 2017), but around 20 per cent (in 2005 and 2010; Rözer et al. 2016). Moreover, the overall state of private precaution can also be integrated into flood loss estimation

models such as the flood loss estimation model (FLEMO) resulting in more reliable damage estimation at different scales, and, therefore, contributing to more robust risk and vulnerability estimations (Thieken et al., 2008). Hence, understanding and predicting private precaution is essential for future planning and flood risk management (Aerts et al. 2018). This holds for all types of inland flooding, i.e. not only with regard to river floods or pluvial floods, but also with regard to flash floods and rapid surface runoff. These flash floods and rapid surface runoff are currently relatively rare, but due to climate change are potentially

more frequent in Europe in future (e.g. Murawski et al., 2015). However, individual precautionary behaviour it is not yet fully understood, particularly if people are affected by different flood types. Questions must be raised as to whether affected individuals carry out private precautionary measures and what are motivating and demotivating factors for this behaviour. In this context, the protection motivation theory (PMT) (Rogers, 1975) has been frequently used as a psychological model to explain the risk-reducing/protective behaviour of individuals. PMT does so by analysing the influencing factors on coping

strategies and potential positive or negative responses to a risky event. Originally evolved in the health sector, PMT has gained attention in the domain of natural hazards over recent years (Mulilis and Lippa, 1990; Grothmann and Reusswig, 2006; Bubeck et al., 2017). PMT relies on two main cognitive processes - "threat appraisal" and "coping appraisal" - to describe the mental response to a specific threat. Threat appraisal is composed of the perceived consequences and probability of a specific threat. Coping appraisal comprises the variables "self-efficacy" (perception of how well a person is able to carry out protection

measures), "response efficacy" (how effective the measures are believed to be) and "response cost" (the perceived costs in terms of money and effort) (Rogers, 1975; Bubeck et al., 2012).

Here it has been shown that the motivation to protect oneself from flooding cannot be solely explained by risk information, risk perceptions and socioeconomic factors such as income and homeownership (e.g. Baan and Klijn, 2004; Bubeck et al., 2012; Morss et al., 2016) In fact, the PMT results in reliable estimations of protective behaviour due to the combination of

threat appraisal and coping appraisal, where particularly coping appraisal has been evaluated as a good predictor (Floyd et al., 2000; Milne et al., 2000; Bubeck et al., 2012; van Valkengoed et al., 2019). Supportive evidence is given by Hopkins and Warburton (2015), who revealed that flash flood experience among UK citizens does not necessarily lead to higher risk perceptions. Yet, Harries (2012) shows that protective behaviour of flood affected UK citizens is significantly associated with

the perceived probability to be flooded again. Moreover, the potential effects of protective behaviour such as feelings of safety, reduced anxiety, and the fear of uninsurable impacts were also influenced by flood experience. This suggests that apart from risk perception (threat appraisal), different psychological factors influence protective responses which can also result in mal-adaptive behaviour, i.e. ignorance or fatalism.

Further, Bubeck et al. (2018) study flood affected households in Germany and France and identified good social norms and

networks as an important factor for better coping abilities after river floods. In particular, Bubeck et al. (2018) found that perceived self-efficacy and response efficacy increased with the number of neighbours who already implemented flood protection measures. When taken together these results suggest that among the influencing factors on protective behaviour, psychological characteristics such as perceived coping abilities, anxiety, negative mental feelings or perceptions of stress as well as avoidant thoughts might play a significant role.

It became clear that private flood precaution might be influenced more by psychology than it has been taken into account by now, raising questions about psychological impacts of flood events in general. This is usually referred to as mental health. Here it can be expected that, severe river floods and flash floods have strong impacts on mental health of affected residents in addition to financial losses. For instance, Mason et al. (2010) reveal that certain criteria for psychiatric disorders such as the post-traumatic stress disorder (PTSD) as well as high scores of anxiety and depression are met within one quarter to one third

of flood-affected study participants among different communities in the UK. Additionally, an increased exposure to floods may also be connected to negative mental health effects due to the disruption of daily routines, financial loss and situational stress, especially if social support by family and friends is missing (Bei et al., 2013). This effect is further described and underpinned by Hudson et al., (2019), who found that flood experience is connected to a loss in subjective well-being among flood affected residents in Vietnam, while females tend to recover slower than males. This was also found by Bubeck and

Thieken (2018) for Germany. Additional evidence for negative mental health effects after floods is given by Wagner (2007), who suggests that models of anxiety and coping can be related to fears of different hazard types. Those models describe reactions and coping strategies of people who are guided by vigilance (i.e. actively searching for threat-related information) and avoidance (i.e. denial & distraction). Moreover, a comprehensive review of Fernandez et al. (2015) on flood related mental health issues as well as Foudi et al. (2017) strongly support the assumption that, in case of flood exposure, especially water

depth and high flow velocities have a negative impact on mental health in terms of increased levels of PTSD, anxiety, as well as depression. This is also supported by Lamond et al. (2015) who suggest that psychological symptoms such as stress and anxiety remain as a result of severe flooding and flood damage. Further they reveal that mental health issues are related to post-flood mitigation actions, where especially relocating seems to be a suitable coping measure.

Therefore, it is important for an integrated flood risk management to consider how river and flash floods differ and how their occurence might lead to different impacts on individuals. However, only a few studies consider individual psychology, outside of PMT concepts, in flood preparedness decisions although it can be expected that they play an important role. Thus, the aim of this work is to identify patterns of psychological impacts of floods with a focus on differences among people affected by either flash floods or river floods. In the following step, the psychological characteristics are related to the overall protective behaviour. Accordingly, the following hypotheses were raised:

**H1**: Flash floods, in comparison to slow-onset river floods, show a different psychological impact on affected people. Negative effects such as stress and feelings of being helpless are expected to be more pronounced, since flash floods are rough, emerge suddenly and therefore represent an unpredictable danger for health and property.

**H2**: Negative psychological impacts are connected to a lower probability for showing precautionary behaviour because negative feelings might hamper the individual's willingness and self-confidence as well as the overall motivation to implement precaution measures.

**H3**: Psychological indicators such as e.g. the feelings of stress and burden that people still perceive several months after a damaging flood had occurred or self-reported coping abilities can be used as a proxy to explain precautionary behaviour because such mental feelings and attitudes are connected to the motivation and intention to protect oneself in future.

The first hypothesis is based on the evidence given by Gaume et al. (2009), Mason et al. (2010), Lamond et al. (2015) and Foudi et al. (2017) and is tested by comparing psychological characteristics of people which are affected by different flood types and strengths. Thus, groups of similar psychological characteristics (psychological indicators) are created. Then the differences in the distributions of the indicator values, i.e. shifts to lower or higher indicator scores, are assessed for each flood type.

To answer the second and third hypotheses which is based on evidence given by Wagner et al. (2007), Mason et al. (2010), Lamond et al. (2015) and Bubeck et al. (2018), precaution indicators ("planned precaution" and "implemented precaution") are created first. Consequently, all following analyses use the planned precaution as dependent variable and psychological indicators as independent variables. In the next step, a Bayesian approach and negative binomial regressions are applied and the resulting probability distributions of conditional variable dependences as well as regression coefficients are evaluated. Negative binomial regression can be used to model ordinal count data where variance and mean are not equivalent. The Bayesian approach has been frequently used in psychology (e.g. Wetzels et al., 2011), medicine and other disciplines. The Bayesian approach can assess data uncertainty which is particularly helpful for studies that rely on relatively small data sets. This is achieved by allowing prior information independent of the data can be included (Van de Schoot et al., 2015). Since this study relies on small data sets, using the Bayesian approach as a supportive analysis helps to interpret main results. In revealing data and model uncertainties, the reliability of future prediction models that are based on these data sets can be evaluated in advance. Accordingly, this study considers Bayesian inference as a method to assess variable relations, that are based on

conditional probabilities and related uncertainties. Preliminary assumptions such as e.g. linear variable relations are therefore not required. Bayesian statistics were also chosen due to the fact that the method enables prior knowledge to be taken into account from other studies that use similar Bayesian approaches. However, to assess the potential direction of the predictor and response variable correlation, the Bayesian approach is supported by a negative binomial regression model. The implementation of all methods is addressed in the next section.

In the end, gaining insights into the psychological impacts of river floods and flash floods and related precautionary behaviour is important for the following reasons:

- The outcome might be beneficial for information campaigns that better support flood affected individuals in different flood prone regions. Various mental coping approaches could also be addressed in such campaigns, since they may vary among different flood types and affected regions. The motivation to implement suitable private flood precautionary measures could be strengthened according to the individual needs of affected people (e.g. Harries et al., 2008; Morss et al., 2016).

- A good understanding of psychology and precaution motivation might result in a model, which indicates the probability for a good precautionary outcome and could be integrated into flood loss modelling and dynamic risk assessments as suggested by Aerts et al. (2018).

- A better understanding of this connection might help to improve future vulnerability and risk estimations and may facilitate the use of alternative data sources to estimate the state of individual precaution. For example, data from online surveys, social media and communication platforms offers a lot of potential to assess individual mental coping strategies such as evasive behaviour or the active remembering after severe events.

These points will be further discussed together with the results of this study in section 3. A further outlook on this topic is given in the conclusion (section 4).

## 2 Data and methods

In this section, the used data is presented and the applied data pre-processing as well as the methodology are explained.

### 2.1 Description of the river flood and flash flood datasets

The individual datasets consist of standard surveys with residents affected by either the river flood of 2013 or the heavy rainfalls in 2016 that led to strong flash floods in some cases. The river flood of 2013 and the flash floods of 2016 are considered for comparison since the two events were very different in terms of the flood dynamics. Additionally, both events were relevant at the national scale. It should be noted that the regions which were affected by the river floods in 2013 and flash floods in 2016 do not overlap. The floods in 2013 were mainly recorded in Saxony and Saxony-Anhalt along the Elbe river and partly

in Bavaria along the Danube river. The heavy rainfalls in 2016 mainly led to flash floods and pluvial floods in Bavaria, Baden-Wurttemberg and North Rhine-Westphalia. .Given the fact that both surveys cover two different flood types, the time lag between the two surveys, i.e. three years, is not expected to cause any effect on the following analyses.

Both surveys were conducted around nine months after the damaging event and were implemented in a similar way, i.e. as computer-aided telephone interviews. The underlying questionnaire of both surveys is identical regarding all questions that were chosen for this study. In general, the questionnaires were designed to improve flood damage estimation of affected households and the assessment of damage driving factors. Hence, the biggest part comprised questions about the flood impact at the affected property, socio-economic characteristics (e.g. age, gender, social status, income, education, homeownership), characteristics of the housing unit (e.g. number of stories or floor space, construction year, number of persons per unit, housing area) and different dimensions of private precaution (e.g. whether certain single protection measures were already implemented before the damaging event or at the time of the interview or are planned to be implemented in the near future), warning and emergency measures (see Thieken et al. 2005; 2017). Yet, various psychological characteristics that address elements of the protection motivation theory (threat appraisal and coping appraisal) as well as avoidance, memories of the event, optimism and further questions about the mental well-being were also recorded. These were – combined with questions about the actual and intended private precaution (see above and section 2.4) – used as the database for this study. An exhaustive list of the analysed psychological variables is given in Table 1. All psychological variable ratings were transformed to follow a self-reported rating scheme of 1 (not once/I do not agree/very low) to 6(7) (few times a day/I fully agree/very high), which ensures their comparability. In this context, four out of nine variable ratings were reversed (see Table 1).

The dataset of the 2013 river flood comprises 1652 responses in total, the 2016 flash flood 601 cases with similar distributions of age (average 59 years) and gender. This study considers only homeowners for all consecutive analyses, since homeowners – unlike tenants – suffer from flood damage on the building itself to a greater extent and also hold a greater flexibility to take potential protective actions (e.g. Grothmann and Reusswig, 2006). The proportion of homeowners within the river flood and flash flood dataset is 82% and 86% respectively, lowering the valid responses to 1366 (2013-flood) and 517 (2016-floods). More information about the samples (type of housing, age, education, and gender) can be found in the Appendix, Table A1.

*(Table 1)*

## 2.2 Separation of weak floods and strong flash floods

In May and June 2016, several places in Germany were hit by flash floods or surface water flooding that differed, however, in intensity and dynamics. In many cases, the heavy rainfall only led to an increased surface water runoff in the vicinity of affected buildings and/or the water entering the basement due to the fact that the sewer system could not cope with the water volume; this is also referred to as pluvial flooding. Yet, in some municipalities, entire villages (such as Braunsbach and Simbach am Inn) were suffering from enormous flash floods and debris flows with strong flow velocities and a very high suspension of

debris – even large rocks – vigorously damaging buildings and infrastructure (Laudan et al., 2017). Therefore, it is crucial to separate rainfall events that led to weak floods or strong flash floods before comparing the psychological impacts among each other and to the 2013 river flood.

The flash flood strength was assessed on the municipal scale. It can be assumed that impressions and effects of the flash flood severity are not particularly dependent on the intensity at the individual house but are rather influenced by the overall appearance and effects of the flood within the village, which also includes impacts on neighbours, friends and infrastructure. It makes sense that mental coping, especially after strong flash floods, is not solely influenced by the individually experienced damage but dependent on broader impressions.

Moreover, not only the impact, but also the potential to be harmed outdoor in case of sudden and strong flow forces may influence the mental coping in regions which can experience strong flash floods. In this context, Morrs et al. (2016) showed that people who perceive flash floods as a risk to their life tend to protect themselves if they receive a flash flood warning. Therefore it can be assumed that the mental impacts after a severe event are differing with regard to the severity within an affected area (e.g. Bei et al., 2013). Consequently, we split the data of the 2016-flood into cases related to pluvial flooding (further refered to as "weak flash floods") and cases that experienced strong flash floods.

The approach to assess the flash flood strength comprises quantitative and qualitative methods and makes use of rainfall data and press articles to evaluate inundation depths and flow velocities. Here, the "Deutscher Wetterdienst" (DWD) provides public download services where precipitation data can be accessed online at cdc.dwd.de/portal/. Accordingly, hourly rainfall data that is based on station observations was downloaded for the days with known heavy rainfalls in May and June 2016. By definition of DWD, a severe weather alert is issued for a particular region if the local rainfall is expected to exceed 25 mm per hour. Thus, if the rainfall exceeded 25 mm per hour at a gauging station, the region was marked to be potentially affected by a severe weather event. An intersection with the survey data was possible since the approximate address of each affected household is provided in the data set. In the next step, an online literature and press article review was conducted for each affected area to find evidence of the flood's intensity. This procedure can be described as a rather qualitative approach. According to the reported damage, impressions of photos and the level of media attention as well as associated rainfall in the area at the particular time based on data from DWD, the surveyed households were classified to weak flash floods (covering pluvial floods) if a low impact in terms of damage and severity was noticed), to medium flash floods (if the impact was considered to between low and high or strong flash floods if a high flood impact could be assumed. For the analysis, only weak and strong flash floods among homeowners were considered and all medium impacts were excluded to reduce the noise of poorly classified data and to increase the effect of flash flood intensity. The count of cases for weak flash floods is n = 293 and for strong flash floods n = 116.

## 2.3 Defining main psychological indicators

In this study, indicators and single variables are defined as follows. A single variable describes the outcome of a standalone question within the survey (also called item). An indicator stands for a newly created variable that consists of two or more

single variables or items. To answer the first hypothesis, four main psychological indicators were considered; threat appraisal, coping appraisal, burden and evasion. The indicators were created by combining variables according to what is described in the literature, i.e. Creamer et al. (2003). This is because the literature suggests that combining relevant items such as e.g. "I had trouble staying asleep" and "Any reminders brought back feelings about it" can create indicators such as e.g. "intrusion",

that reduce information to the core content. Furthermore, Grothmann and Reusswig (2006) and Bubeck et al. (2012) describe the items that constitute the factors of the PMT, which are especially relevant as the main psychological indicators. Subsequently, the four main indicators used in this study are defined as "threat appraisal", "coping appraisal", "burden" and "evasion", which show low intercorrelations and offer a certain comparability to other studies such as Bubeck et al., (2017) for example. The four indicators are thus defined and created as follows.

Threat appraisal and coping appraisal are defined according to the PMT (see Section 1 Introduction) Threat appraisal consists of the perceived probability of being affected again by a flood event and the perceived impact of such a future event. Coping appraisal comprises self-efficacy, response efficacy and response cost. Burden describes a measure for the negative psychological load of the individual experience and can be used to reveal differences among residents affected by different flood types. Burden is measured via the responses to the questions regarding "often thinking of the event" and "stress still

today". Evasion comprises the responses to the variables "avoidance" and "fatalism" (see Table 1). We argue that Evasion can be seen as a measure of the effort to get the experience of a damaging flood out of one's mind in order to cope with the threat, which is regarded as mal-adaptive behaviour in the PMT.

Burden and evasion were developed by following the general procedure in psychology surveys to combine expressive psychological items (e.g. Ware and Sherbourne, 1992; Kroenke et al., 2001) and taking high correlations among psychological

variables into account. Accordingly, the creation of the indicators burden and evasion required pre-processing of the data, correlation tests and the evaluation of preliminary results. Thus the preliminary results are shortly presented in this section.

The rank correlations among the single psychological items were assessed using ordination plots (principle component analysis) and correlation tables (Spearman's Rho, corrected after Holm (1979), done in R Studio 1.1.414 (R version > 3.5.0), using the package "psych" version 1.8.12 and "pwr" version 1.2-2). According to the tests, subjective stress which is still felt

at the time of the interview and the frequency of remembrance of the event show a correlation of 0.54 for weak flash floods (complete cases n=279, lower / upper 95% confidence interval [0.39, 0.66]), 0.46 for strong flash floods (n=115, [0.19, 0.66]) and 0.50 for river floods (n=1152, [0.42, 0.56]) with a p value of <0.05 in all cases. Furthermore, avoidance and fatalistic thoughts reveal a correlation of 0.23 for weak flash floods (n=275, p<0.05, [0.05, 0.40]), 0.29 for strong flash floods (n=113, p=0.07, [-0.01, 0.53]) and 0.18 for river floods (n=1242, p<0.05, [0.09, 0.26]). Therefore, we combined avoidance and fatalistic

thoughts as two different strategies of mal-adaptive behaviour. See the Appendix for the correlation tables (Figures A1, A2 and A3).

Based on these results, the subjective stress still felt at the time of the interview and the frequency of remembrance was combined to the indicator burden, while avoidance and fatalistic thoughts constitute the indicator evasion. In this context,

burden describes the degree of negative psychological load that is still perceived at the time of interview and evasion resembles mal-adaptive behaviour, e.g. trying to supress the experience.

The distributions of threat appraisal, coping appraisal, burden and evasion were further analysed using the Dunn's Test, which is based on the non-parametric Kruskal-Wallis rank sum test results. These tests are suitable for assessing the differences among the distributions of ordinal-scaled data, which does not fulfil assumptions of normality and equality of variance. Here, the Kruskal-Wallis rank sum test calculates discrepancies among the rank sums of all values within the compared indicators. The derived Kruskal-Wallis statistic is then compared to the expected average difference among the sum of ranks via Dunn's Test. Similar to a power analysis, the effect size and significance are revealed for a given sample size. The outcome represents a measure for the disparity and shift of compared distributions. This approach reveals significant differences in psychological impacts which were predominantly caused by weak flash floods, strong flash floods and river floods (see section 3.1).

## 2.4 Indicator on precautionary behaviour

Since both datasets (flash floods/heavy rainfalls and river floods) include questions about the current and intended state of flood precaution (see Section 2.1), indicators can be derived that each reflect the actual or intended state of precaution. In total, 16 different private precaution measures were considered in the questionnaire. For each private precaution measure, individuals were asked to mark them as "implemented before the event", "implemented after the event", "will be implemented in the next six months", or "not intended to be implemented". The 16 measures reflect different types of precautionary behaviour. First, they comprise the collection of information about flood protection and flood risk as well as information within seminars; in this study these three items were combined to one measure "collecting information". Further, insurance, contributing to neighbourhood networks, flood-adapted story usage, flood-adapted interiors, relocating heating and electricity, securing heat and oil tanks and improving the flood safety of the building were considered as individual measures. Finally, a combined variable on "water barriers" that consist of installing backflow prevention and/or installing water barriers as well as a combined variable "emergency preparation" which consists of having no noxious liquids in the cellar, installing pumps, having generators available and/or anticipatory planning of supplies. This information was further used to create an aggregated indicator on precautionary behaviour. This was done on the basis of the studies of Kreibich et al. (2005), Thieken et al. (2005) and Büchele et al. (2006). Kreibich et al. (2005) compared the flood damage mitigation potential of different private precaution measures among German households that were affected by the severe river flood in 2002. The study revealed that flood adapted use, a better interior fitting and the relocation of heat and electrical utilities lower the damage ratio of buildings by 46%, 53% and 36% respectively (Kreibich et al., 2005). Related studies based on the same data also proposed to combine different private precaution measures to an index; weights can be assigned to reflect the measure's damage reducing potential (Thieken et al., 2005, Büchele et al., 2006). For the index, all measures implemented by a pre-defined point in time were summed up and weighted. Thereby, the measures "flood-adapted story usage" and "flood-adapted interiors" were weighted ten times due to their high damage-reducing potential (see Kreibich et al. 2005). Further, the items "relocating heating and electricity", "securing heat and oil tanks", "improving the flood safety of the building", "water barriers", and "emergency preparation"

were weighted five times. In summary, the indicator based on the weighted single measures reflects a certain state of precaution, which also includes the relative effectiveness to lower building and content damage. The procedure results in a precaution indicator with values ranging from 0 to 48, which are further reclassified into values ranging from 0 (low precaution) to 8 (high precaution), based on equal interval sizes. In this study, this approach was applied once to measures that were already implemented at the time of the event, and – for a second time - to measures that were implemented after the event and that are intended to be implemented shortly (up to 6 months after the time of the interview). In the results and discussion section (section 3.2.), the two indicators reflecting the state of precaution at two different points in time are compared to provide a better understating of the data set.

In this context, an important aspect to consider is the fact that people who had already implemented many effective precaution measures when the flood occurred - and thus reflect a good level of private precaution - can be expected to show a lower motivation for further measure implementation (saturation effect). Therefore, cases are disregarded if the count of already implemented measures or missing answers is equal or exceeds the half of the overall measure count of 16 measures (>= 8), since it is hardly possible to obtain meaningful results for the planned precaution in such cases. Hereby, it is also ensured that there is no bias towards low precaution motivation in the subsequent analysis, which is caused by an already high precaution level prior to the event.

## 2.5 The Bayesian approach

Bayesian statistics can be applied to calculate probability distributions from a limited set of observations and to quantify related uncertainties. The statistical model takes prior knowledge into account ($P_0$(model parameter), also called prior) and assesses the likelihood to observe the data, if specific model parameters are given (P(data|model parameter)). This results in a probability density for the model parameters, conditioned on specific data (P(model parameter|data), also called posterior) (Puga et al., 2015). The underlying principle is Bayes theorem, which leads to the following equation (1):

$$P(model\ parameter|data) \sim P(data|model\ parameter) * P_0(model\ parameter) \tag{1}$$

The likelihood P(data|model parameter) is based on the binomial distribution for each response variable (planned precaution) and predictor variable value. The binomial distribution was chosen due to the fact that it provides probability estimations solely on the occurrence and non-occurrence of two possible values, as given in the dataset. E.g. it can be used to model the probability density of flipping a coin (one side of a fair coin occurs in 50% of all cases). The binomial distribution is thus defined as (2):

$$P(k\mid p,n) = \binom{n}{k} * p^k * (1-p)^{n-k} \tag{2}$$

- n = count of specific predictor variable value

- k = count of specific response variable value, given n

Here, the estimated parameter (p) resembles the specific combination probability of two variable values. More precisely, it indicates the likeliness to observe a specific response variable value, if a specific predictor variable value is given. To our
knowledge, no similar studies exist which are based on comparable datasets and equal psychological indicators, thus, no prior knowledge is taken into account in this study. This means that the prior, which influences the estimation of the parameter (p), was chosen to be uniformly distributed on [0, 1]. Eventually, the Bayesian analysis results in posterior distributions that indicate the conditional probability density of the occurrence of two variable values. This means that a given variable value (e.g. value 3 out of 6 possible values among planned precaution) occurs with a particular value of another variable (e.g. value 6 out of 6
possible values among avoidance) to most likely e.g. 45 per cent (peak of parameter p).

## 2.6 Average posterior distributions, Jensen-Shannon divergence and regression tests

In order to test the second and third hypotheses, the psychological indicators as well as the single psychological variables (see Table 1) were analysed with regard to their connection to the planned precaution indicator, using the Bayesian approach, the Jensen-Shannon divergence (JSD) and a negative binomial regression model. Both, the psychological indicators and the single
variables were analysed to reveal differences between the general procedure in psychology to combine similar items/variables and studying all variables separately.

First, the planned precaution indicator was used as the dependent variable and all posterior distributions for each psychological indicator (coping appraisal, threat appraisal, burden and evasion) as well as all single psychological variables (see section 2.5.) were calculated according to the Bayesian approach, excluding all non-existent combinations. Next, the posterior distributions
were combined per variable by applying the weighted arithmetic mean (Figure 1). In detail, this means that the combined posterior distribution shows the likeliness of the likeliness of all mutually occurring variable values in a single graph. Here the distribution shape of parameter p (i.e. its highest peak) resembles the most likely probability of mutual occurrence, given the dataset. Yet, it is not specified which variable values occur mutually. In the next step, a weighted arithmetic mean posterior is calculated by randomising the analysed variable to obtain a random occurrence of predictor and response variable. This step
is necessary to get the particular reference posterior shape, which is exclusively influenced by the distribution of the predictor and response variable.

*(Figure 1)*

The combined, weighted posterior distributions were further evaluated using the JSD. JSD is based on Shannon Entropy (H) which is defined by (3):

$$H(X) = -\sum_{i=1}^{n} p(x_i) \, log_2\big(p(x_i)\big) \qquad\qquad (3)$$

- p = probability mass function
- X = discrete variable with possible values {$x_1,\ldots,x_n$}

Hence, the JSD can be expressed by the more known Kullback-Leibler divergence (which gives the same information) and is defined by (4):

$$JSD(P,R) = H\big(0.5 * (P + R)\big) - 0.5\big(H(P) + H(R)\big) \tag{4}$$

- P = posterior distribution
- R = reference posterior distribution

This divergence represents the degree of mutual information between two or more variable distributions and the strength of their connection (or to which degree they are distinguishable). Consequently, the JSD was used to assess the similarity of each posterior distribution and its reference posterior distribution to reveal if they differ from each other. The JSD can take any value between 0 and 1. If the JSD of the reference posterior and the calculated posterior is 0, both underlying variables (e.g. the planned precaution indicator and burden) are independent from each other and do not show any relation apart from random effects. If the JSD is greater than 0 however, these variables show a certain information gain if one is explained by the other. If the JSD is 1, both underlying variables are identical.

Complementary to the Bayesian approach (i.e. the combined posterior distributions and divergence), negative binomial regressions were performed for each flood type, using the planned precaution indicator as response variable and the psychological indicators as well as the single psychological variables as predictors. The negative binomial regression was chosen due to the fact that the "planned precaution" indicator consists of ordinal discrete (count) values which are restricted between 1 and 8 and follow an overdispersed (variance is greater than the mean) Poisson distribution (tested in R 1.1.414, using the packages "logspline" and "fitdistrplus"). Since the posterior distributions and divergence computations are solely based on probabilities, information gain and prediction applicability can be assessed. Yet it is not clear how both variables relate to each other (i.e. positively or negatively). Thus it is supported by a negative binomial regression model which indicates significant positive or negative correlations of variables with the "planned precaution" indicator.

## 3 Results and discussion

In this section, the differences in the distributions of the psychological indicators are presented and discussed first. In the next step, the implemented and planned precaution indicators are presented and compared before the indicators and single psychological variables are analysed by evaluating the posterior distributions, the JSD and regression coefficients. Subsequently, the hypotheses are discussed at the end of this section.

### 3.1 Psychological indicator distributions

Figure 2 illustrates the frequency distributions of the four psychological indicators, i.e. coping appraisal, threat appraisal, burden and evasion, and also includes the Dunn's Test results.

*(Figure 2)*

Regarding coping appraisal (Figure 2, top left), the indicator distributions and Dunn's Test reveal significant differences between strong flash floods, river floods and weak flash floods. People affected by strong flash floods show generally lower ratings than people who suffered from weak flash floods or river floods. Still, most of the respondents reported medium coping
appraisal ratings (Figure 2, top left).

The results indicate that people who were affected by strong floods feel generally less able to cope with the situation and the implementation of protective measures. Although the effects are not strongly pronounced, a significant difference to weaker flash floods (pluvial floods) becomes apparent which might be due to the different (potential) flood impacts. In this context, our data of the 2016 flood shows higher structural damage on building substance for strong flash floods (38 per cent, 13 per
cent and 2 per cent smaller cracks, bigger cracks and collapsed elements respectively) than for weaker events (25 per cent, 4 per cent and 1 per cent smaller cracks, bigger cracks and collapsed elements respectively). A similar outcome is indicated when comparing the difference between strong flash floods and river floods. However, the results are not significant which might be due to the fact that different flood processes are covered by the 2013 data set. Although it has not been tested whether a lack of awareness regarding precautionary strategies, missing protection information campaigns or other effects lead to a
lower coping appraisal for strong flash floods in general, the effects could also be explained by the fact that people do not believe in a high efficiency of precaution measures in case of strong flash floods.

Concerning threat appraisal, the significantly lower ratings of people affected by strong flash floods are remarkable, since it could be assumed that severe and damaging events lead to stronger feelings of threat in the first place (Figure 2, top right). Yet, these results could be explained by the fact that people who were affected by strong flash floods believe similar events to
be very unlikely to recur in near future, resulting in lower feelings of threat. Although Hopkins and Warburton (2015) showed that flash flood experience does not necessarily lead to higher risk perceptions, it is unknown, to which degree lower feelings of threat are caused by a lower flash flood experience itself. Since almost all surveyed households experienced a strong flash flood for the first time (82%), they may not believe that they will be affected again. However, an analysis of the threat appraisal with corrected data in terms of flood experience (considering just households that experienced a flood for the first time) reveals
a similar picture, i.e. threat appraisal is significantly lower for people who were affected by a strong flash flood in comparison to people who were affected by weak flash floods and river floods (see Appendix, Figure A4). This again supports the findings of Hopkins and Warburton (2015).

Still, e.g. Murawski et al. (2015) have shown that there may be an increase in severe flash floods in regions, which were formerly not perceived as flash flood-prone. This further highlights the importance of specific information campaigns in this context to counteract mal-adaptive behaviour. Weak flash floods and river floods show a relatively similar distribution (not significantly distinct from each other) with a peak at medium threat appraisal ratings and a peak at the highest threat appraisal
rating. This might be due to the weaker nature of the flash flood event and the higher perceived probability to be affected by a similar event again. With regard to river floods, a number of people in Germany have been affected more than three times within a relatively short period between 2002 and 2013, which might also contribute to a pronounced feeling of threat in residents who have been affected by river floods. This is in line with Mason et al. (2010), who find that the fear of reoccurrence of a flood event and anxiety is increased with repeated experience of damaging events.

The ratings of burden are significantly lower for people affected by weak flash floods, indicating a lower psychological load and feelings of stress (Figure 2, bottom left). The distributions of strong flash floods and river floods are on the other hand shifted to higher ratings of burden. This clearly illustrates the connection between the "severity" of an event and the resulting negative psychological impacts, which is in line with Mason et al. (2010) and Bei et al. (2013), who reported that a greater impact in terms of daily routine disruption, financial loss and evacuation is associated with significantly worse effects on
mental health. In contrast to the "severity" of an event, the type of the event (flash flood or river flood) does not seem to have an effect on burden, since strong flash floods and river floods do not display any significant distribution differences (Figure 2, bottom left).

Similarly, the indicator evasion shows a significant difference in the distributions only with regard to weak flash floods (Figure 2, bottom right). This could be explained by the same effect that weak events or events leading to less severe impacts in general
result in less pronounced feelings of avoidance and fatalism (i.e. in less mal-adaptive behaviour). Here, evasion especially differs between people affected by weak flash floods and river floods. One reason could be the comparatively high frequency and severity of river floods in Germany which could lead to evasive behaviour of repeatedly affected residents. In fact, evasive behaviour can be described as a particular (but mal-adaptive) strategy to cope with severe events, enabling affected individuals to emotionally distance themselves from oppressive situations, as described by Mason et al. (2010).

In summary, the indicators are particularly insightful in case of strong flash floods. The combination of feeling less able to cope with such an event as well as a low perceived threat but yet an increased burden means that people feel an emotional pressure and do not see efficient ways to deal with the situation on their own. A comforting thought then might be the assumption that a similar event will not happen again soon. It can be expected that this leads to mal-adaptive behaviour, although damages on buildings after rapid and strong flood events are usually high. This is a contrast to weaker flood events
and river floods, where risk communication, insurance and private precaution measures are more established. These results again highlight the importance of information campaigns in regions potentially affected by strong flash floods.

**3.2 Precaution indicators**

Since the "planned precaution" indicator is used as response variable within all further analyses its distribution will be presented first in this section. Furthermore, the planned and shortly realised precaution (in the following called planned precaution) is compared to the precaution implemented before the event (in the following called already implemented precaution) (Figure 3).

*(Figure 3)*

By evaluating the distributions of already implemented precaution measures (Figure 3, left side) and planned precaution (Figure 3, right side) it becomes apparent that people who have been affected by river floods show slightly higher scores of already implemented precaution measures. This is in line with Kienzler et al. (2015) and Spekkers et al. (2017). Regarding weak and strong flash floods, the score of already implemented precaution measures is considerably low while it can be noticed that the planned precaution scores are relatively low for all flood types. Especially in the case of river floods, affected people reveal a low motivation for (further) precaution in future. This is also true for people who were flooded the first time.

15 **3.3 Posterior distributions and regressions of the psychological indicators**

Aim of these analyses is to reveal, if psychological indicators or single psychological variables show an influence on the planned precaution. In general, the posterior distributions and regression results are based on a low number of data points, especially in the case of weak and strong flash floods (see Table 2, N). Yet, the results indicate certain positive and negative connections of the psychological indicators to the planned precaution indicator.

*(Figure 4)*

The weighted arithmetic means of all posterior distributions reveal in general a wide range of likely probabilities for the conditional dependence of variable ratings. In the case of weak flash floods for example, it is second most likely (second highest posterior peak) that a particular burden rating is always reported together with a specific rating of the planned precaution to 52 per cent (most likely to 9 per cent due to the highest posterior peak at this point). For coping appraisal, the most likely percentage would be 7 per cent. For threat appraisal and evasion, the most likely percentages are 10 and 19 per cent, respectively (Figure 4, top left). Other posterior peaks are however visible, yet less likely. As mentioned in section 2.6., the posterior shapes are greatly influenced by the distribution of the predictor and response variables. Since the planned precaution indicator is Poisson-distributed with the highest value counts among the lowest ratings, similar posterior shapes can be found in all cases with peaks around 10% and 50%. Yet, considering the reference posterior for burden (Figure 4, top left), the highest JSD is revealed for burden, respectively (Figure 5). The JSD for coping appraisal, threat appraisal and evasion

however is low for weak flash floods. Additionally, the regression results indicate a significant positive relationship of burden and the planned precaution for weak flash floods (Table 2). It can be concluded that, if anything, burden is the most significant and useful indicator to predict the planned precaution among all indicators. Here, perceived feelings of burden are positively related to a higher precaution motivation. This result is in line with Lindell et al. (2009) and Lamond et al. (2015), who find

that often thinking and talking about a hazardous event as well as mental health issues are positively correlated with the intention to adapt to the hazard. Our results confirm that this might also be the case for flooding.

The posterior peaks of strong flash floods are less pronounced which is due to the small dataset of 76 observations (Figure 4, top right & Table 2). In this case, a pattern is observable in which again burden and evasion show distributions slightly shifted to higher probabilities. Yet, the most likely coherence of the psychological indicators and the planned precaution is between

14% and 22% for strong flash floods. Regarding the JSD, evasion reveals a certain information gain when describing the planned precaution, yet the effect is relatively weak (Figure 5). Simultaneously, evasion does not show any significant linear relationship with the planned precaution (Table 2). Thus, a distinct nonlinear pattern among the variables can be expected with regard to this dataset. All other indicators show almost no divergence and no information gain. According to the regression results, burden reveals a slightly negative coherence in this case, yet the significance level is only between 0.1 and 0.05. In

general, the results of the strong flash flood analysis should be interpreted with caution due to the low number of observations. Concerning river floods, all psychological indicators show a peak around 50, up to 60 per cent and a relatively similar posterior shape that is caused by the distribution of the planned precaution indicator (Figure 4, bottom). In the case of burden, a posterior peak at 69 per cent is recognizable, which is remarkably different from the reference posterior shape. Accordingly, the JSD reveals a pronounced information gain for burden, while coping appraisal, threat appraisal and evasion reveal weak divergences

(Figure 5). Yet, the regression results reveal only slight positive and negative coherences for the significant variables burden and threat appraisal (Table 2). These facts speak for a distinct, assumingly nonlinear coherence pattern for burden and the planned precaution, while the other psychological indicators show no significant information gain. However, similar to weak flash floods, stronger feelings of burden seem to result a higher protection motivation, which is again in line with Lindell et al. (2009) and Lamond et al. (2015).

The results contradict studies that found a high relevance of coping appraisals for precautionary behaviour to a certain degree, which claim that higher coping appraisals are connected to preparation and precaution intentions (Floyd et al., 2000; Milne et al., 2000; Bubeck et al., 2012; van Valkengoed et al., 2019). Thus, better insights into the factors of PMT and the actual connection to intended precaution as well as longitudinal studies are needed with regard to flooding. Further, the PMT could potentially be expanded with relevant variables that cover mental health and feelings.

*(Figure 5)*

*(Table 2)*

### 3.4. Rankings and regressions of single psychological variables

Figure 6 shows the JSD of the single psychological variables for weak flash floods, strong flash floods and river floods, indicating the information gain with regard to the planned precaution. In contrast to most of the other variables, the high divergence for "often thinking of the event" is remarkable for weak flash floods and river floods. Only for river floods, a relatively high JSD can be seen with regard to "response efficacy", "response cost" and "fatalism". Compared to Figure 5, it has to be concluded that variables which make up the indicators usually do not show an equal JSD. This is especially true for "often thinking of the event" and "stress still today", which constitute burden. Here, "often thinking of the event" seems to be decisive for high values of burden. In the case of evasion for strong flash floods, however, a combination of the respective variables fatalism and avoidance leads to a higher information gain. The variables that constitute threat appraisal, namely "fear of severe effects again" and "believe in being affected again" do not show any information gain, (Figure 6), which is also reflected in Figure 5.

*(Figure 6)*

Furthermore, the regression results of the single variables indicate almost no significant relationships with the planned precaution indicator (Table 3). Regarding weak flash floods, "often thinking of the event" is significantly connected to a higher planned precaution while for strong flash floods, "fatalism" reveals a significant negative connection. In the case of river floods, no variables are significant (Table 3).

*(Table 3)*

When comparing the analysis of the psychological indicators and the single variables, it can be summarised that a combination of items, as it is practised by e.g. Ware and Sherbourne (1992) and Bei et al. (2013), does not lead to more consistent and meaningful results in this case which is mainly reflected by similar JSDs. Moreover, the regression models of the single variables (Table 3) reveal a higher explanation power ($R^2$), especially in the case of weak flash floods, highlighting the importance of particular single psychological items. So the question remains, which method is the most suitable to combine variables. In this study, only few psychological items/variables were available while surveys to assess mental health comprise various indicators with up to 22 items (e.g. Ware and Sherbourne, 1992; Bei et al., 2013). By combining items, the inconsistencies among reported answers can be lowered and the predictive validity of indicators can be raised, facilitating the creation of psychological profiles (Ware and Sherbourne, 1992; Creamer et al., 2003). The analysis in this study follows this idea and indicates a certain importance of basic psychological indicators or variables for the motivation to implement precaution measures in future. However, the surveys, which are used in this study, primarily focus on direct damage and explanatory variables (see Thieken et al., 2017) and hence only comprise few significant questions which do not necessarily

follow the established scheme of psychological surveys such as for example the 36-Item Short Form Survey (SF36), which is widely used to monitor the quality of life of patients. It has to be noted that more meaningful outcomes may be produced by more standardised questions and items that have been validated in psychology. Within follow-up studies that rely on surveys, adjusting and adding questions should be considered for better psychological assessments.

## 3.4 Discussion of the hypotheses

> *H1: Flash floods, in comparison to slow-onset river floods, show a different psychological impact on affected people. Negative effects such as stress and feelings of being helpless are expected to be more pronounced, since flash floods are rough, emerge suddenly and therefore represent an unpredictable danger for health and property.*

According to Figure 2, it is not the flood type but the strength/severity of the flood induces negative psychological effects. Among strong flash floods and river floods, no significant difference in stress becomes apparent except for threat appraisal where the distribution of strong flash floods is based on a relatively small dataset of 76 records (Figure 2, top right). Yet, this difference could be explained by the fact that the perceived threat of a strong flash flood event is lower due to the severity and the uniqueness of the event itself. Affected people perceive a (future) strong flash flood event as being less likely than people who have been affected by river floods. Thus, future disaster risk management in Germany may also take into account that the individual threat perceptions of affected residents may be low. Therefore, information campaigns in flash flood prone regions should be promoted, especially if various studies suggest an increase in severe flash flood events due to climate change and a change in weather patterns such as strong precipitation events (e.g. Murawski et al., 2015). However, since all remaining burdensome and negative psychological effects vary with regard to the flood severity and do not significantly vary among different flood types, the first hypothesis must be rejected.

> *H2: Negative psychological impacts are connected to a lower probability for showing precautionary behaviour because negative feelings might hamper the individual's willingness and self-confidence as well as the overall motivation to implement precaution measures.*

The most surprising result is that a high level of burden increases the protection motivation instead of affecting it negatively (Figure 5 & Table 2). However, otherwise no strong connections between strong psychological impacts and planned precaution were found. This may be explained by two reasons. First, the assessment methods of psychological items as well as the items themselves do not follow established psychological assessment routines or surveys, what potentially decreases data consistency and accuracy. Second, subtle effects on precautionary behaviour that are caused by psychological aspects may be covered by incidental effects, due to the small sample sizes. This is particularly true for strong flash floods, leading to high uncertainties. However, it is revealed that the indicator burden and, from a general point of view, thinking often of the event - which may not go hand in hand with negative feelings - as well as the subjective stress are slightly positively connected to the precaution motivation among different flood hazards. This is contrary to the hypothesis but yet a valuable result, indicating a certain motivation of affected residents to protect themselves even after a severe and burdensome flood event. Here, the perceived

"recency" and presence of the event may play a role in preparedness decisions. This result further supports Bei et al., (2013), who reported that affected people with worse mental and physical health show a higher willingness for coping strategies. However, since negative psychological impacts are slightly positively connected to the precaution motivation, the second hypothesis must be rejected.

*H3: Psychological indicators such as e.g. the feelings of stress and burden that people still perceive several months after a damaging flood had occurred or self-reported coping abilities can be used as a proxy to explain precautionary behaviour because such mental feelings and attitudes are connected to the motivation and intention to protect oneself in future.*

According to the correlation results, weak coherences (JSDs) as well as high uncertainties, the identified psychological
indicators are mainly not suitable for explaining precautionary behaviour (see Figure 4, Figure 5, Table 2 & Table 3). Here it is remarkable that this result contradicts studies on PMT which confirm a positive relation of high coping appraisals and the willingness to implement precaution measures (Floyd et al., 2000; Milne et al., 2000; Bubeck et al., 2012; van Valkengoed et al., 2019). In this context, it should be taken into account that the time lag between, and the perceived priority of measure implementation after a damaging event is differing among affected people. Hence, more detailed information is needed
concerning the temporal dynamics of planned or implemented precaution measures which might lead to clearer results. As already mentioned, further reliable insights could be obtained by applying standardized and established surveys to assess psychological characteristics. In this regard, Creamer et al. (2003), for example, confirm the usefulness of the Impact of Event Scale - Revised (IES-R). This scale is a widely used item-based survey that measures traumatic stress in order to assess symptoms of the post-traumatic stress disorder (PTSD). However, they also find that the main factors of the IES-R, i.e.
"hyperarousal", "avoidance" and "intrusion" do not provide a good account of the data due to correlations among single items and suggest the use of fewer, or more diversely composed, factors/indicators.

A very diverse and promising future field might also be the application of data mining techniques and the use of alternative data sources to increase data amounts, to facilitate the psychological profiling and predicting precautionary behaviour by different methods. An issue of telephone surveys is that the data is becoming biased towards older participants when based on
landlines (Greenberg and Weiner 2014). Alternatively, by implementing and making use of online surveys, smartphone applications and contracts with companies, larger amounts of valuable data could be collected accounting for people from all age groups. For further use, algorithms such as Neural Networks or deep learning algorithms may be applied to this data to create or categorize psychological aspects such as the expected level of burden or evasion in case of an event. Those techniques might result in good predictions of psychological behaviour and the connected precaution motivation and can theoretically be
transferred to other regions but yet imply certain challenges. Firstly, large amounts of consistent and high quality data have to be collected on the condition that data security and personal rights are considered. Secondly, the interpretation of results in terms of causality and meaning is hampered due to the black box character of the analysis, even though potential results might

show a certain robustness. In this context, established routines of mental assessments have the advantage of a better transparency and will continue to play an important role in future.

Eventually, a lot of research still has to be done in that regard. This study, however, reveals that stronger feelings of stress and often thinking of an event (i.e. the perceived burden) are connected to a higher precaution motivation, although the usability
as a strong predictor within probabilistic models is limited due to the weak effect strengths. Thus, the third hypothesis can only be partly confirmed.

## 4 Conclusion

The aim of this study was to investigate psychological impacts in flood affected residents that are caused by different flood types as well as the influence of these impacts on precaution motivation. Furthermore, the usefulness of psychological
indicators and individual psychological variables to predict precautionary behaviour was evaluated. In this context, four psychological indicators and a precautionary motivation indicator were created and differences in psychological impacts among flood types were analysed by using the Kruskal-Wallis rank sum test and Dunn's Test. The connection of these indicators and the individual variables to the precaution motivation was assessed by applying negative binomial regressions and Bayesian statistics as well as evaluating the posterior distributions using the JSD.
The study shows that **generally it is not the flood type, but rather the overall severity of a flood event** leads to stronger mental impacts among affected individuals. The exception is threat appraisal, where people affected by strong flash floods report lower values. Here it is indicated that people are under emotional pressure but do not know how to cope with the situation and probably do not believe in a good efficiency of private precaution measures. In terms of mental coping they rely on a lower probability for such an extreme event again, all potentially leading to mal-adaptive behaviour.
In general, strong flash floods and river floods result in higher values for the indicators burden and evasion when compared to weak flash floods. The examination of psychological variables reveals that **potentially useful indicators of planned precaution, such as burden, can be derived**. Here it is revealed that **people who report a higher mental load (indicator: burden) indicate a higher motivation to implement private precaution measures in future, hinting the importance that mental health might have for precaution.** Yet, the overall strength of different variable connections and the predictive power
are generally low, which may be partly due to small sample sizes. Additionally, the fact that the planned precaution is heavily biased towards low values, i.e. generally not intended in future among all flood types impairs the clarity of results. When combining psychological variables, or items to derive a more robust indicator of mental health, established procedures which are applied in pure psychological studies should be taken into account. Considering the surveys which are used in this study, the predictive validity can, potentially, be enhanced by combining items based on more specific and standardised questions on
mental health may lead to more robust results. Therefore, standardised psychological assessments should be considered within follow-up studies.

Overall it is indicated that, in particular, **the frequency of remembering an event is positively connected to preparedness intentions**. Therefore, recommendations for disaster assistance and risk communication are difficult to derive, especially with regard to increase the protection motivation of flood-affected individuals and helping with the individual recovery. This could be achieved by strenghtening the beliefes in the effectiveness and applicability of precautionary measures, informing about the

risk and offering mental support. For heavy rainfalls that lead to pluvial floods as well as for river floods, examples on precaution from the neighborhood could be communicated in combination with risk maps for specific areas. Regarding strong flash floods it could be meaningful to include affected people in strategies that can be realised on municipality level (e.g. retention areas), highlighting the dangers of such events and informing about specific private precaution measures that could mitigate lower building damage.

In terms of future development and regarding psychological assessments that are based on publicly available information, further research may also focus on comparisons to established mental health surveys and validity checks to gain knowledge about the usefulness of alternative data sources for predicting individual behaviour. With the help of advanced intelligent learning algorithms (e.g. random forests, neural networks and deep learning), psychological profiles could thus be created. Those might be used to develop sophisticated models and predict the state of precaution in areas which have not been flooded

recently, all based on data given voluntarily by residents. Surveys that capture the state of precaution are still an alternative option. After all, further research is required to estimate the predictive power of different psychological models which rely on mental health assessments and aim to quantify protective behaviour in the context of flooding.

**Data availability**

Part of the data that support the findings of this study are openly available at https://cdc.dwd.de/portal/.

**Competing interests**

The authors declare that they have no conflict of interest.

**Acknowledgements**

This work was developed within the framework of the Research Training Group "Natural Hazards and Risks in a Changing World" (NatRiskChange; GRK 2043/1) funded by the Deutsche Forschungsgemeinschaft (DFG). The surveys were conducted

as a joint venture between the GeoForschungsZentrum Potsdam, the Deutsche Rückversicherung AG, Düsseldorf, and the University of Potsdam. The survey on the 2013-flood was conducted in the framework of the research project "Flood 2013" funded by the German Ministry of Education and Research (BMBF; funding contract 13N13017). All survey data are owned by the authors.

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

**Table 1: List and explanation of the psychological variables used in this study.**

| Variable | Original variable scale | Original question or the statement (shortened), for which the degree of agreement was asked |
|---|---|---|
| Believe in being affected again | 1 (I fully agree)… 6 (I do not agree) | **Statement**: It is likely to be affected again by a flood event. |
| Fear of severe effects again | 1 (I fully agree)… 6 (I do not agree) | **Statement**: A future flood event will not be as bad as the recent event. |
| Self-efficacy | 1 (I fully agree)… 6 (I do not agree) | **Statement**: I personally do not feel able to implement at least one private precaution measure. |
| Response efficacy | 1 (I fully agree)… 6 (I do not agree) | **Statement**: Private precaution measures can reduce the flood damage. |
| Response cost | 1 (I fully agree)… 6 (I do not agree) | **Statement**: Private precaution measures are too expensive. |
| Stress still today | 1 (no stress)… 6 (high stress) | **Question**: How much do you currently feel stress and negative emotions caused by the flood event? |
| Often thinking of the event | 1 (not once)… 7 (few times a day) | **Question**: How often have you thought about the event within the last six months? |
| Avoidance | 1 (I fully agree)… 6 (I do not agree) | **Statement**: I do not like to think of future flood events. |
| Fatalism | 1 (I fully agree)… 6 (I do not agree) | **Statement**: One is in general helpless regarding future flood events and the damage. |

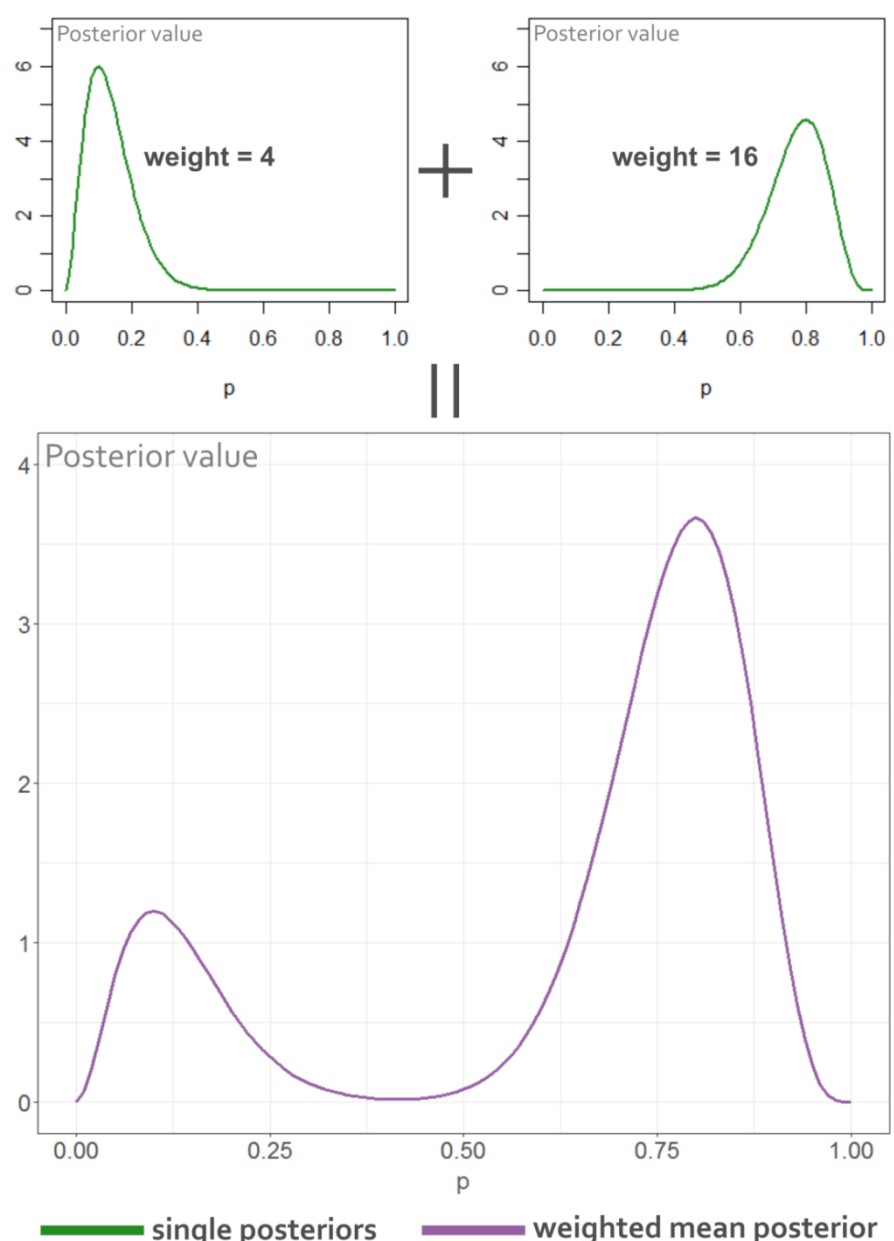

**Figure 1: Example graphic explaining the creation of the weighted arithmetic mean posterior. The double peaks are a result of the combination of all posteriors that are calculated for each variable combination. The posteriors are weighted according to the sum of occurrences within the dataset. In this case the weighted mean posterior means that, given the example dataset of 20 data points, it is most likely that a specific predictor variable rating occurs together with only one specific response variable rating to 80%.**

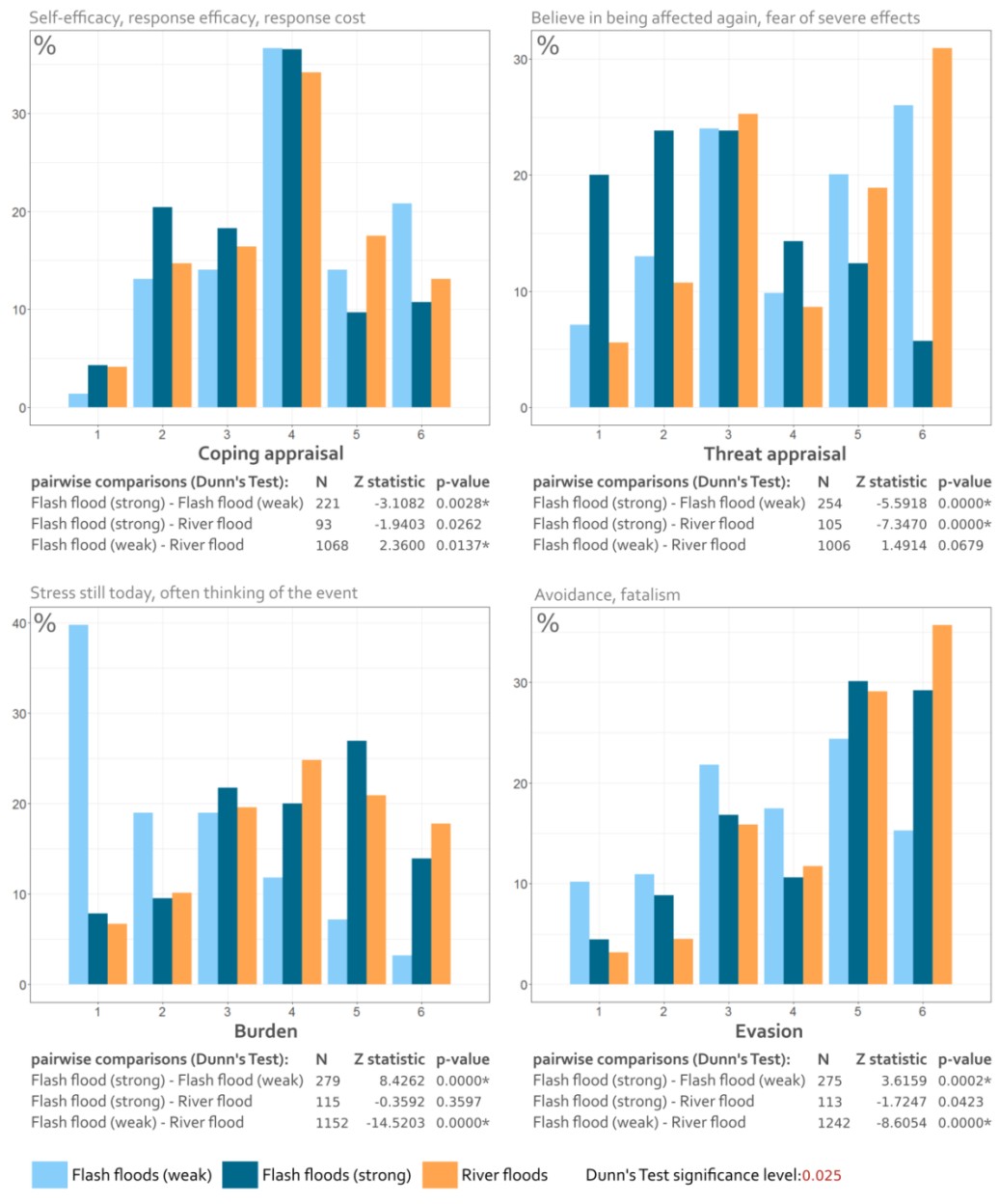

**Figure 2: Relative distributions of the combined psychological indicators for each flood type and Dunn's Test results. The results of the Dunn's Test reveal the direction shift of each distribution compared to the other distributions (negative means a shift towards lower values, positive a shift towards higher values), by also indicating the strength and significance of the shift (Z-statistic and p-value).**

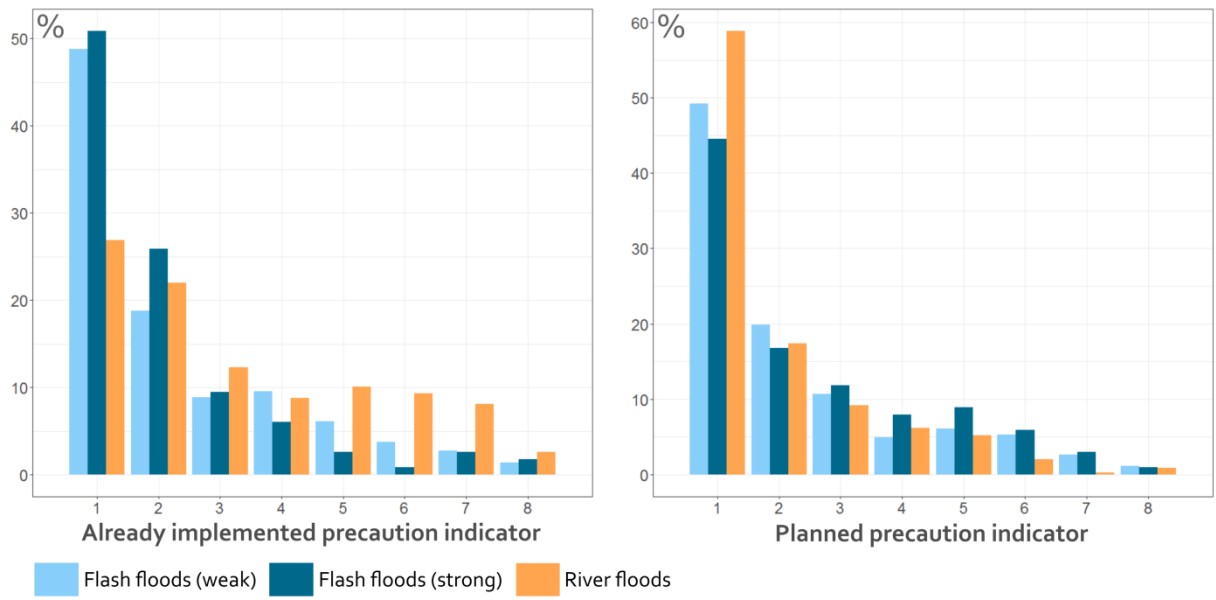

**Figure 3: Relative distribution of the already implemented precaution indicator (left) and the planned precaution indicator (right) for weak flash floods (n=293), strong flash floods (n=116) and river floods (n=1366). The X axis represents the implementation of, or the intention to implement effective precaution measures as described in section 2.4. The higher the value, the more effective measures have been implemented, or will be implemented in near future.**

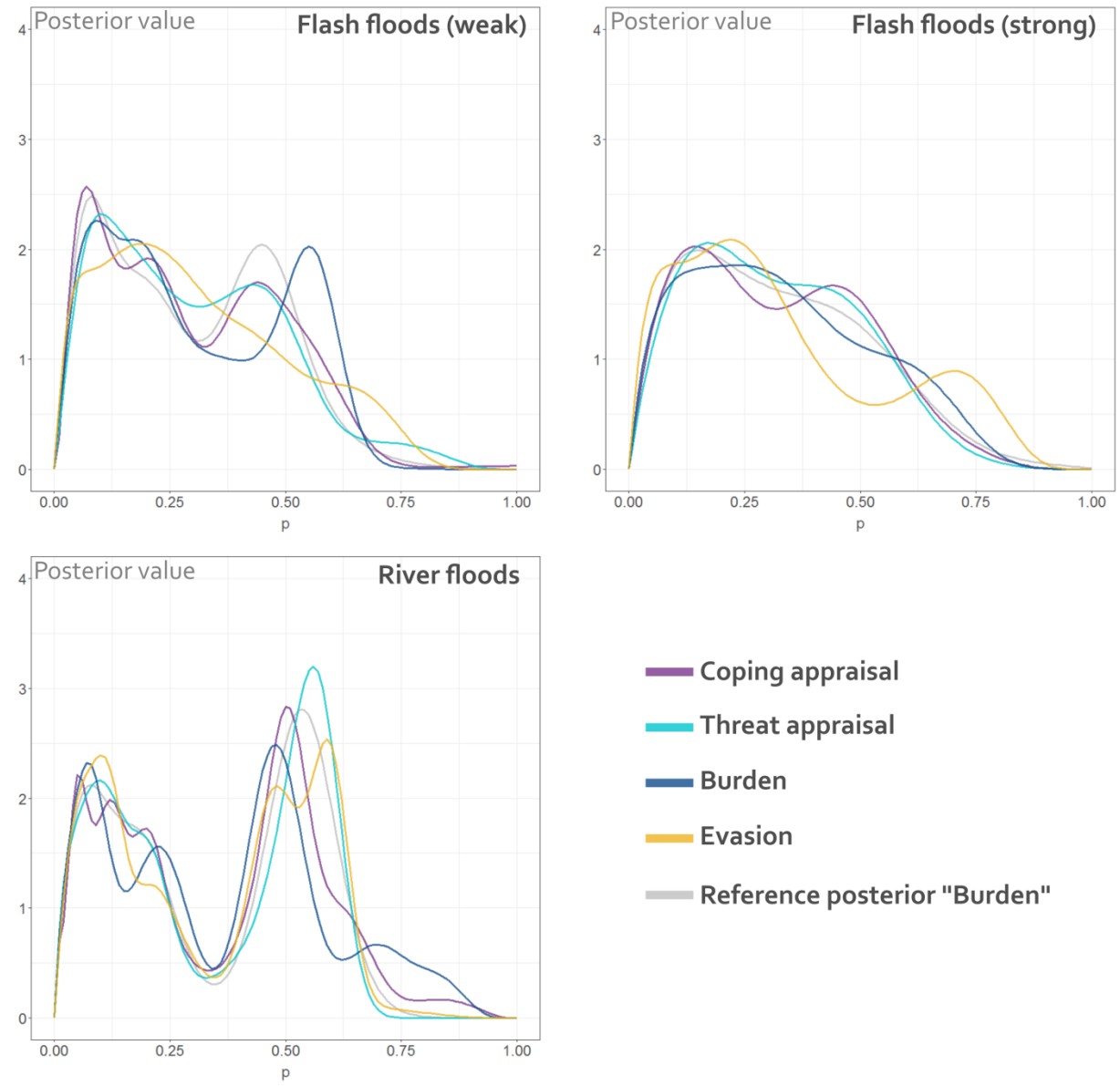

**Figure 4: Weighted arithmetic mean of all posterior distributions for the psychological indicators "Coping appraisal", "Threat appraisal", "Burden" and "Evasion", given weak flash floods (top left) strong flash floods (top right) and river floods (bottom left). The reference posterior is shown for "Burden" only.**

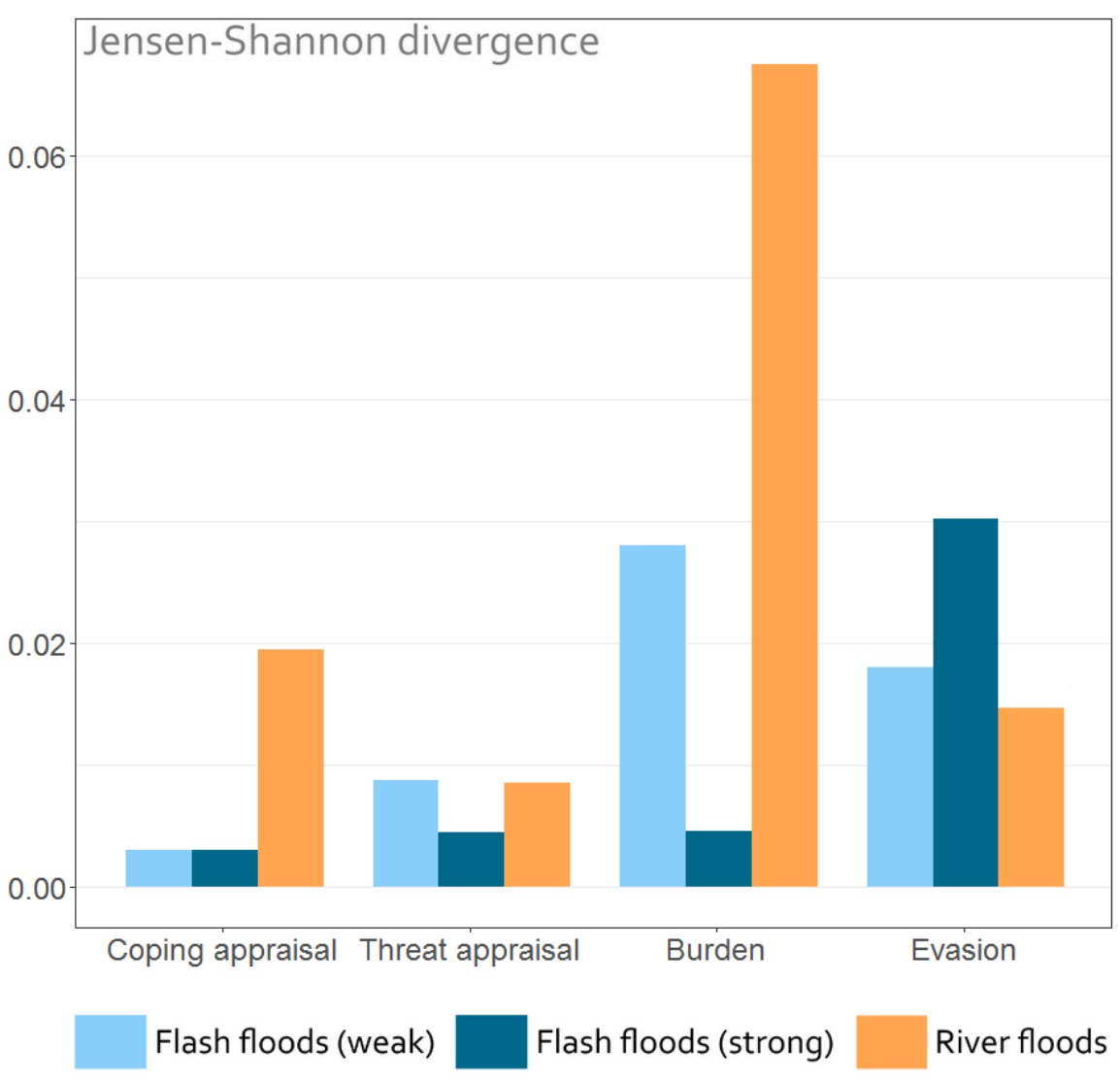

**Figure 5: Jensen-Shannon divergence ranking of the psychological indicators. Higher values indicate a higher information gain, if the planned precaution is explained through the particular indicator.**

**Table 2: Coefficients of the negative binomial logistic regression models for weak flash floods, strong flash floods and river floods with the psychological indicators as predictor variables and the "planned precaution" indicator as response variable.**

| Predictor variable | Flash floods (weak) | Flash floods (strong) | River floods |
|---|---|---|---|
| *Intercept* | 0.673 * | 1.585 ** | 0.483 * |
| Coping appraisal | 0.012 | 0.011 | 0.024 |
| Threat appraisal | -0.013 | -0.016 | -0.038 ' |
| Burden | 0.134 *** | -0.105 ' | 0.054 * |
| Evasion | -0.024 | -0.059 | 0.020 |
| AIC | 667.26 | 293.01 | 1422.30 |
| R² | 0.08 ** | 0.06 | 0.03 * |
| N | 177 | 76 | 419 |

Note: 'p-value <.10, *p-value <.05, **p-value <.01, ***p-value <.001.

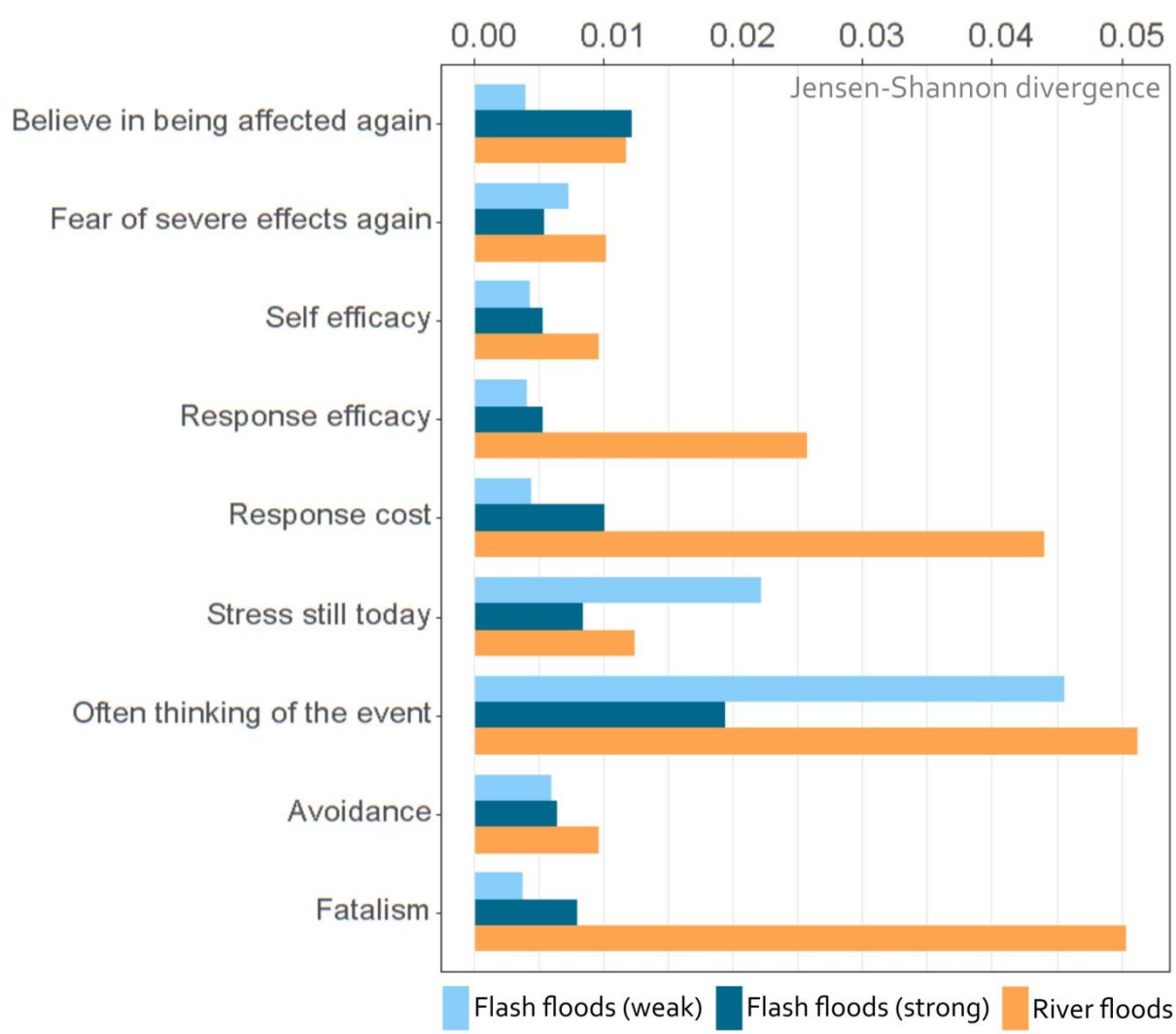

**Figure 6: Jensen-Shannon divergence ranking of single psychological variables. Higher values indicate a higher information gain, if the planned precaution is explained through the particular variable.**

**Table 3: Coefficients of the negative binomial logistic regression models for weak flash floods, strong flash floods and river floods with the individual psychological variables as predictor variables and the "planned precaution" indicator as response variable.**

| Predictor variable | Flash floods (weak) | Flash floods (strong) | River floods |
|---|---|---|---|
| *Intercept* | 0.619 ' | 1.644 ** | 0.510 ' |
| Believe in being affected again | -0.031 | 0.032 | -0.028 |
| Fear of severe effects again | 0.002 | -0.024 | -0.020 |
| Self-efficacy | -0.003 | 0.002 | -0.007 |
| Response efficacy | 0.042 | -0.019 | 0.027 |
| Response cost | -0.017 | 0.006 | -0.002 |
| Stress still today | 0.040 | -0.056 | 0.036 |
| Often thinking of the event | 0.102 ** | -0.047 | 0.022 |
| Avoidance | -0.044 | 0.030 | 0.012 |
| Fatalism | 0.020 | -0.103 * | 0.009 |
| AIC | 669.34 | 300.24 | 1429.10 |
| R² | 0.12 ** | 0.10 | 0.04 |
| N | 177 | 76 | 419 |

Note: 'p-value <.10, *p-value <.05, **p-value <.01, ***p-value <.001.

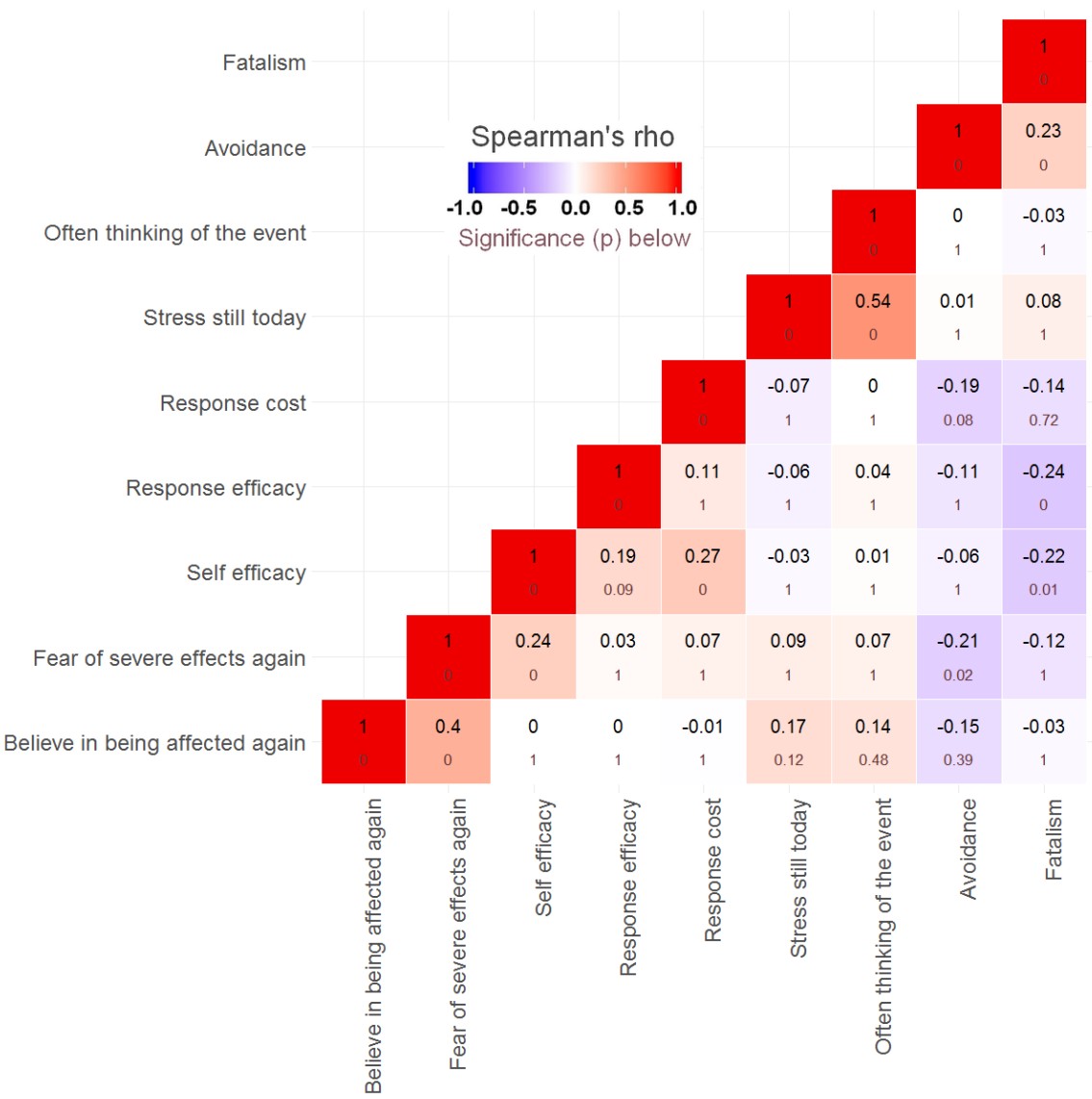

**Fig. A1: Correlation table of single psychological variables for weak flash floods.**

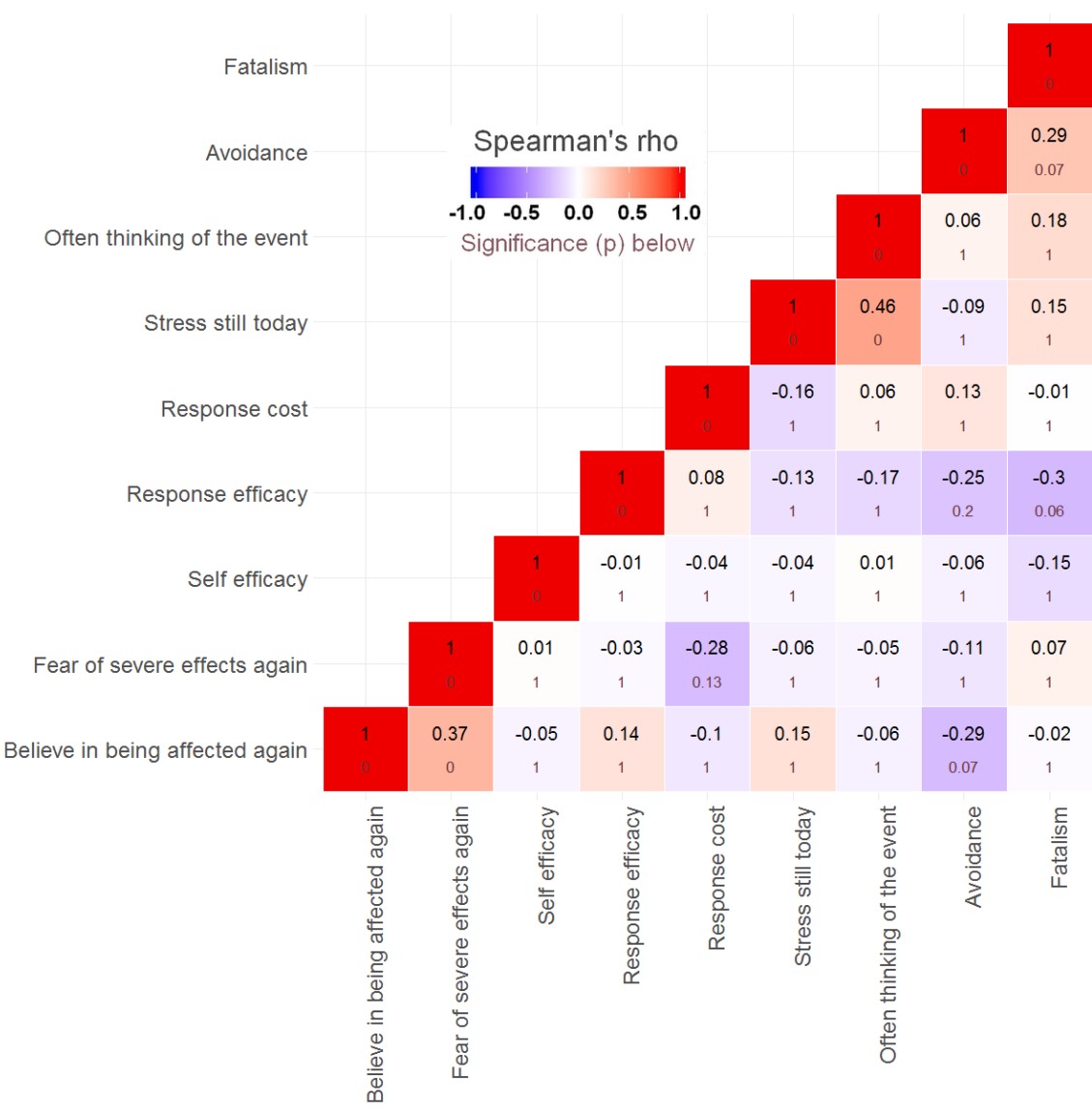

**Fig. A2: Correlation table of single psychological variables for strong flash floods.**

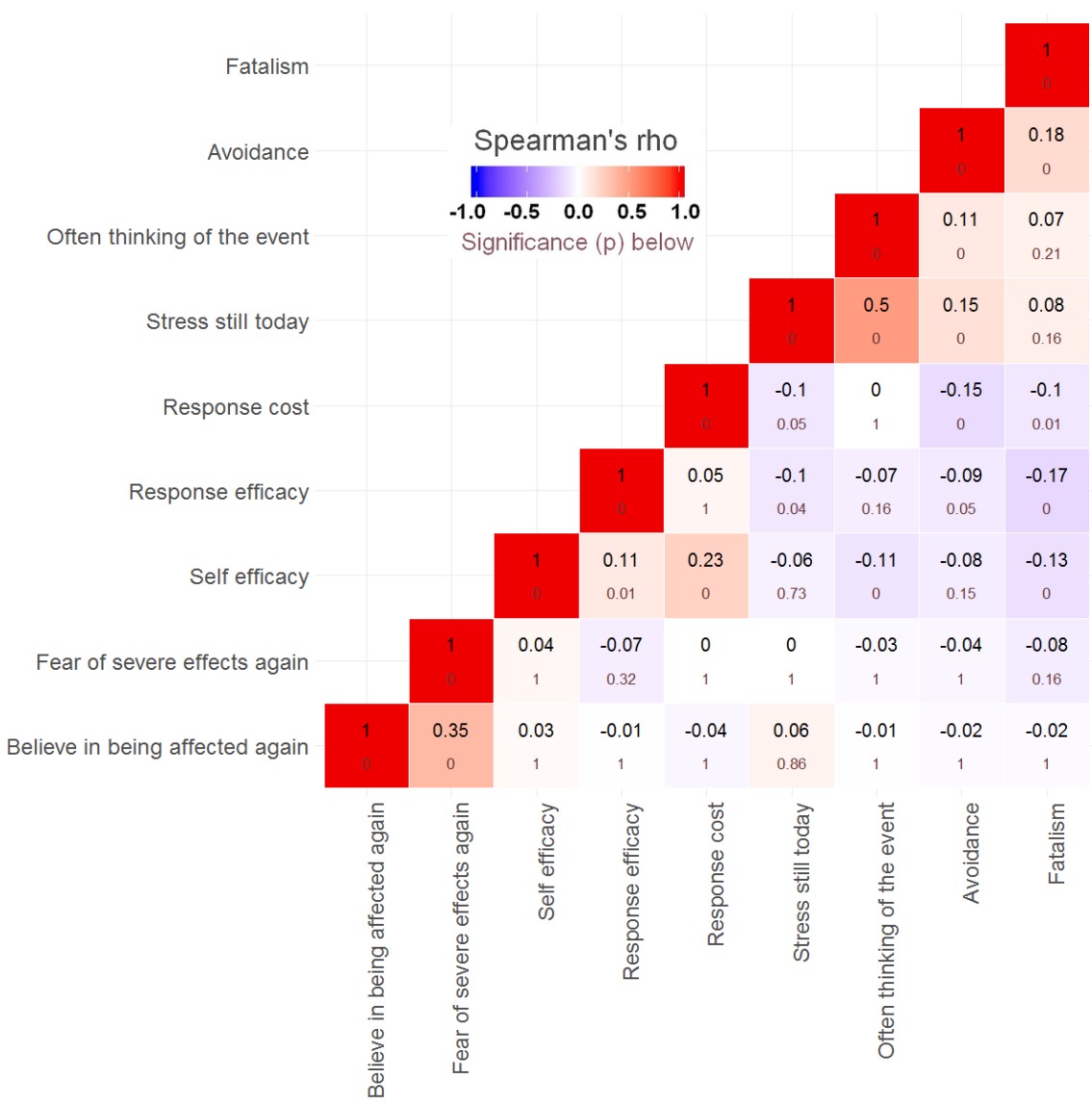

**Fig. A3: Correlation table of single psychological variables for river floods.**

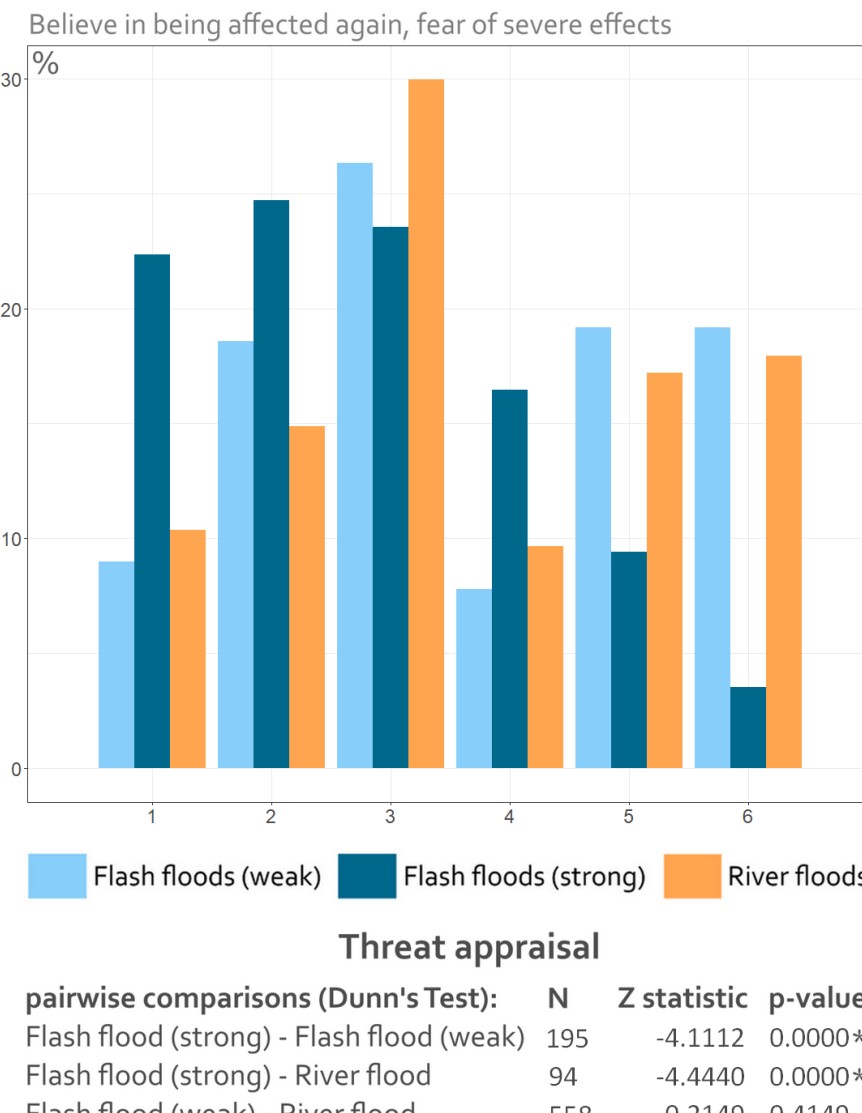

Believe in being affected again, fear of severe effects

**Threat appraisal**

| pairwise comparisons (Dunn's Test): | N | Z statistic | p-value |
|---|---|---|---|
| Flash flood (strong) - Flash flood (weak) | 195 | -4.1112 | 0.0000* |
| Flash flood (strong) - River flood | 94 | -4.4440 | 0.0000* |
| Flash flood (weak) - River flood | 558 | 0.2149 | 0.4149 |

**Fig. A4: Relative distribution of Threat appraisal among each flood type and Dunn's Test results. The data was corrected for flood experience, i.e. all households which only experienced a flood once. The results of the Dunn's Test reveal the direction shift of each distribution compared to the other distributions (negative means a shift towards lower values, positive a shift towards higher values), by also indicating the strength and significance of the shift (Z-statistic and p-value).**

**Table A1: Information about the samples and datasets**

| Variable | Flash flood dataset 2016 (n=517) | River flood dataset 2013 (n=1366 ) |
|---|---|---|
| | n | n |
| **Type of housing** | | |
| Single-family house/duplex house | 293 | 778 |
| Semi-detached houses | 45 | 124 |
| Terraced houses | 50 | 116 |
| Farm houses | 17 | 72 |
| Other | 16 | 18 |
| NA | 96 | 258 |
| **Age of the respondents [years]** | | |
| 16-30 | 20 | 31 |
| 31-50 | 104 | 281 |
| 51-70 | 257 | 642 |
| >70 | 99 | 280 |
| NA | 37 | 132 |
| **Education** | | |
| No school graduation | 3 | 13 |
| Secondary modern school | 82 | 289 |
| Middle school/apprenticeship | 200 | 483 |
| AVCE (Vocational Certificate of Education)/technical diploma | 35 | 82 |
| University degree | 164 | 419 |
| NA | 33 | 80 |
| **Gender** | | |
| Male | 229 | 581 |
| Female | 288 | 785 |

