# Peer review of "Flash floods versus river floods – a comparison of psychological impacts and implications for precautionary behaviour"

_Natural Hazards and Earth System Sciences, 2018_

## Referee Comment (RC1) · Tim Harries (Referee) · 8 May 2019

The comment was uploaded in the form of a supplement:
https://www.nat-hazards-earth-syst-sci-discuss.net/nhess-2018-407/nhess-2018-407-RC1-supplement.pdf

---

## Referee Comment (RC2) · Anonymous Referee #2 · 19 May 2019

Review: Flash floods versus river floods – a comparison of psychological impacts and implications for precautionary behaviour

This paper tries to find a relationship between psychological indicators and the use of precautionary measures. The paper has a very clear structure and methodology, which makes it easy to follow on an abstract level. The main problem with the paper is that for someone not very familiar with all the statistical methods the details of the paper are difficult to follow. I therefore recommend major revisions because the work needs a lot of clarification.

A general limitation of the study seems to be that the people who experienced river

floods experienced them multiple times in the last 10 years. While the people who experienced flash floods seem to have fewer past experiences. Is there a possibility that this frequency of past experiences may be a stronger signal than the flood type? Is there a way to correct your data for the number of flood experiences people have had?

Specific comments: Page 4, line 8-20, these are some very technical sentences, could you explain your approach in a more intuitive way and introduce the technical methods later. Currently this is difficult to read without prior knowledge about the statistical methods that are applied.

Page 4 line 24-30: Could you sketch in a bit more detail how you see this being used in the future. We don't know these psychological indicators for everyone when we make a damage model. It might even be easier to ask directly about precautionary measures than to assess their psychology. Using social media information as proxy might be a solution but I like to see these arguments made a bit more thoughtful and if that's the way to apply it I like to see that back in the discussion and maybe a recommendation to study how social media clues can be linked to the indicators used in this paper. You mention several times the "protection motivation theory", please give a brief explanation of this, you can't assume all your readers know about this.

Page 7, line 5, please first explain what burden and evasion are before explaining the motivation behind it.

Give a proper explanation of Kruskal-Wallis rank sum test, Dunn's Test, the Jensen-Shannon divergence and regression tests directly after you first mention these methods. Maybe don't mention them too early in the text. Give both an intuitive and a brief mathematical explanation of the methods.

2.5 Explain why you use Bayesian statistics, you now just jump into the explanation without first motivating the choice.

[Figure]

2.5 Why did you choose to use Bayesian statistics if no prior is available? What is the advantage of using Bayesian statistics?

Figure 2: why is threat appraisal lowest for strong flashfloods? Does it make sense that if something extreme happened to you, you feel the probability that it will happen again to be lower? (your argument on page 11, line 10). Maybe threat appraisal is lower because they only experienced it once while the river floods and weak flash floods were experienced more frequently. If however you would go to another region where only one weak flash flood or river flood was experienced these results may look very different. You should probably discuss that limitation in the study.

Figure 4: This figure is not very intuitive can you explain a bit more what the reader sees here.

Figure 4: Why do you see the double peaks in the probability distributions?

Page 14, line 17: You say this is common practice in psychology, can you provide a reference for this?

Page 15, H2: I think the findings make much more sense than the hypothesis.

---

## Author Comment (AC1) · 3 Aug 2019

**Response to Referee #1, Tim Harries RC1: nhess-2018-407-RC1, 2019**

Author: Jonas Laudan[1]
Co-authors: Gert Zöller[2], Annegret H. Thieken[1]

[1]University of Potsdam, Institute of Environmental Science and Geography, Karl-Liebknecht-Strasse 24-25, 14476 Potsdam, Germany
[2]University of Potsdam, Institute of Mathematics, Karl-Liebknecht-Strasse 24–25, 14476 Potsdam, Germany

We thank the reviewer for the helpful and constructive comments as well as the reasonable suggestions to improve the paper. Thus, we would like to follow many of the reviewer's hints, addressing all helpful comments.

15

**Comments of the reviewer**

**Main point 1:**

The clarity of the argument is often poor and the logic sometimes flawed/missing

20 • E.g. P3.29: "This suggests that…" Why does it? Surely there are other, alternative explanations? E.g. the social norms/networks mentioned earlier on the same page. What is the evidence/theory to suggest that psych factors might vary between flood types? Given that this is the main hypothesis of the paper, it requires careful justification.

**Answer 1:**

25 Thank you for the hint, we will elaborate more on the explanations, why psychological characteristics might have strong influences on the flood protection motivation and why they potentially vary among flood types. Suggested changes in the text:

"It has further been shown that the motivation to protect oneself from flooding cannot be solely explained by risk information, risk perceptions and socioeconomic factors such as income and homeownership (e.g. Baan and Klijn, 30 2004; Bubeck et al., 2012; Morss et al., 2016). Supportive evidence is given by Hopkins and Warburton (2015), who revealed that flash flood experience among UK citizens does not necessarily lead to higher risk perceptions. Yet, Harries (2012) shows that protective behaviour of flood affected UK citizens is significantly associated with the perceived probability to be flooded again while potential effects of protective behaviour such as feelings of safety;

anxiety, and the fear of uninsurable impacts are influenced by flood experience. Having analysed flood affected
households in Germany and France, Bubeck et al. (2018) identified good social norms and networks as an important
factor for better coping abilities after river floods. Especially the trust in its own abilities and the belief in a good
measure effectiveness increase with the number of neighbours, who already implemented flood protection measures.
Eventually, these results suggest that among influencing factors on protective behaviour, psychological characteristics
might play a significant role."

(…)

"River floods usually occur after long-lasting rainfall or snowmelt within large catchment areas and result in slow-
rising water levels. In contrast to river floods, flash floods emerge within (small) catchments where slopes are steep and
defined, resulting in unpredictable flow dynamics that can be rough in terms of a high sediment transport, high flow
velocities and forceful discharge (Borga et al., 2014). The forecast of such flood events is not yet reliable since they can
develop with very short lead time. Apart from potentially high damage on buildings and infrastructure, flash floods can
also cause serious injuries and fatalities (Gaume et al., 2009). Therefore it can be assumed that flash floods are
perceived as a threat for personal health and property and induce negative psychological responses in flood experienced
people.

 After all, only few studies consider individual
psychology in flood preparedness decisions although it can be expected that they contribute to the knowledge in that
regard."

**Main point 2:**
The paper is not easy to read/understand E.g. (but there are numerous other instances):
   • What is meant by "within the individual bounds of possibility" p2? This needs to be more precisely expressed
   • P3 "In general" is too vague.
   • Long sentences – e.g.11.

**Answer 2:**
We will check the manuscript again and increase the readability/clarity of the text at the suggested locations. We will further
rewrite sentences and use better expressions to follow the reviewer's suggestions. Suggested changes in the text:

65    "As a consequence, the German Act on precautionary flood protection in 2005 (Act to Improve Preventive Flood Protection) requires residents in flood prone areas to undertake appropriate private precautionary actions ."

(…)

"It has been shown that private precaution measures can significantly reduce the mean damage ratio (flood damage in
70    relation to the total building/content value) to households and household contents up to 53 per cent and thus play an important role in comprehensive flood management strategies (Kreibich et al., 2005; Thieken et al., 2008; Merz et al., 2010)"

(…)

"Here, evasion especially differs between people affected by weak flash floods and river floods. One reason could be
75    the comparatively high frequency and severity of river floods in Germany which could lead to evasive behaviour of repeatedly affected residents"

80    **Main point 3 (point 1):**

Methods

• The methodology for classifying flash flood strength is opaque and potentially flawed. Was this done for individual homes or for entire areas? The former would be appropriate, but I can't see how it would be possible using online searches and press reports. The latter would be insufficiently fine-grained, because the intensity of impact often varies
85    dramatically between homes in the same street/area. The authors need to justify their reliance on crude estimates of physical damage for an analysis that looks at psych impacts. Are they assuming a close correlation between the two? If so, they should present citations supporting this.

**Answer 3 (point 1):**

90    The flash flood strength was assessed on a coarser area since we assume that impressions and effects of the flash flood severity are not particularly dependent on the intensity at the individual house. We rather believe that they are influenced by the overall appearance and effects of the flood within the village, which also includes impacts on neighbours, friends and infrastructure. For example, Bei et al., (2013) describe similar phenomena such as negative mental implications due to the disruption of daily routines with regard to an affected infrastructure. It makes sense that mental coping, especially after
95    strong flash floods, is not solely influenced by the individually experienced damage but dependent on broader impressions.

Moreover, not only the impact, but also the potential to be harmed outside in case of sudden and strong flow forces may influence the mental coping in regions which can experience strong flash floods. Therefore it can be assumed that the mental impacts after a severe event are differing with regard to the severity within an affected area.

In addition, we cross-checked our classification with answers concerning the pathway the water entered the building (data are not shown in the paper). Overall, our classification captures weak flash floods (comprises pluvial flooding), medium flash floods and strong flash floods.

**Main point 3 (points 2-7):**

Methods

• Justification needed for the exclusion of moderate strength floods.

• The authors need to justify including 'fatalistic thoughts' in a group named 'avoidance' p7

• Is it justifiable to include information-gathering in the same category of response as physical adaptation? Some reflection is needed on this issue and the key concept of precautionary behaviour needs to be defined accordingly.

• It would be helpful to include a power analysis.

• It would be good to clearly label the research as 'secondary analysis', rather than leaving it to the reader to deduce this.

• The authors should discuss the implications of the three years that separate the two surveys.

**Answer 3 (points 2-7):**

Thank you for these suggestions. We will clarify the different steps of the analysis and discuss the data with regard to the time between the surveys. Suggested changes in the text:

"Thus, the aim of this work is to identify patterns of psychological impact with a focus on differences among people affected by either flash floods or river floods. In a next step, the psychological characteristics are related to the overall protective behaviour. Accordingly, the following hypotheses were raised:"

(…)

"The regions which were affected by the river floods in 2013 and flash floods in 2016 differ almost completely. Further, apart from an increase in insurance density regarding river floods, no specific developments concerning flood risk management and flood precaution are indicated during these years. Given the fact that both surveys also cover two different flood types, the time lag between the two surveys, i.e. three years, is not expected to cause any effect on the following analysis."

For clarification: The group in which 'fatalistic thoughts' is integrated is not named 'avoidance' but 'Evasion'. See description in paper:

130      "Evasion comprises the variables "avoidance" and "fatalism" and can be seen as a measure for the effort to get the experience of a damaging flood out of one's mind in order to cope with the threat."

We will further define the concept of precautionary behaviour beforehand. Suggested changes in the text:

     "The focus was shifted to a more integrated flood management, where also structural precaution measures (i.e.
135      waterproof sealed cellars i.e. dry-proofing, wet-proofing, relocation of heating and electrical utilities) as well as non-structural flood protection measures (i.e. adapted interior fitting and flood-adapted use such as avoiding water-sensitive furniture in the cellar) became increasingly important (Kienzler et al., 2015; Thieken et al., 2016b; Laudan et al., 2017)."

140 For clarification: The category information-gathering and other adaptation measures are combined within the precaution indicator according to their potential to reduce physical damage. The calculation is based on weights that are assigned to each precaution measure that have been derived from analyses mainly done by Kreibich et al. (2005) and Thieken et al., (2005). See description in paper:

     "Thus, the indicator of already implemented precaution measures and the indicator capturing planned precaution, which
145      is used in this study, consist of single precaution measures that are weighted according to their damage mitigation potential as found in Kreibich et al. (2005), Thieken et al. (2005) and Büchele et al. (2006)."

Regarding the power analysis: We decided to describe the Dunn's test in greater detail instead of including a power analysis. The Dunn's test generally obviates the need for a power analysis and also helps to understand the related figure (figure 2).
150 Suggested changes in the text:

     "The distributions of threat appraisal, coping appraisal, burden and evasion were further analysed using the Dunn's Test, which is based on the non-parametric Kruskal-Wallis rank sum test results. These tests are suitable for assessing the differences among the distributions of ordinal-scaled data, which does not fulfil assumptions of normality and equality of variance. Here, the Kruskal-Wallis rank sum test is preliminary to the Dunn's Test and calculates
155      discrepancies among the rank sums of all values within the compared indicators. The derived Kruskal-Wallis statistic is then compared to the expected average difference among the sum of ranks via Dunn's Test. Similar to a power analysis, the effect size and significance are revealed for a given sample size. The outcome represents a measure for the disparity and shift of compared distributions. This approach reveals significant differences in psychological impacts which were predominantly caused by weak flash floods, strong flash floods and river floods."

160

**Main point 3 (points 8-15):**

Methods

• 6.25: "The indicators are combined according to literature…" Requires more explanation.

• The descriptions of the statistical methods, process and results need to be more accessible to readers not expert in stats or the particular methods used

• References needed for the justification for using Bayesian methods (p4)

• The Bayesian approach 'offers advantages'. The authors need to be specific about what these are. P4

• Clearer and more precise language is needed. E.g. 'the specific variable applicability' – what does this (and the rest of the sentence) mean?

• The surveys were 'equally designed'. This needs to be clearer – were they identical or were there differences?

• It makes no sense to speak of 'an equal distribution of age and gender'. 'Balanced'?

• It would be helpful to have more information about the samples. E.g. response rates; social class/occupation; household types; extent of flood damage – and/or some comparison of the German population to highlight ways in which the samples are/are not representative.

**Answer 3 (points 8-15):**

Thank you for these ideas. We will increase the clarity of our descriptions, justify the applied methods and rewrite certain parts of the text. We will further elaborate the explanation how indicators/items can be combined. Suggested changes in the text:

"The indicators are combined according to literature, i.e. Creamer et al. (2003), who suggest to combine items to create robust indicators. Further, Grothmann and Reusswig (2006) and Bubeck et al. (2012) describe the items that constitute the factors of the PMT, which are especially relevant as main psychological indicators. Subsequently, the four main indicators are defined as "threat appraisal", "coping appraisal", "burden" and "evasion", which also show low intercorrelations and offer a certain comparability to other studies."

(…)

"The Bayesian approach has been frequently used in psychology (e.g. Wetzels et al., 2011) and other disciplines. It assesses the data uncertainty which is particularly helpful among studies that rely on relatively small data sets, while prior information independent of the data can be included (Van de Schoot et al., 2015). Since this study relies on small data sets, using the Bayesian approach as a supportive analysis helps to interpret main results. By revealing data and model uncertainties, the reliability of future prediction models that are based on these data sets can be evaluated in advance. Accordingly, this study considers Bayesian inference as a method to assess variable relations, that are based on conditional probabilities and related uncertainties. Preliminary assumptions such as e.g. linear variable coherences

are therefore not required. "

195 (…)

"The dataset of the 2013 river flood comprises 1652 responses in total, the 2016 flash flood 601 cases with similar distributions of age (average 59 years) and gender."

Additionally, we will include a table in the appendix to provide more information about the samples (age, gender, education,
200 type of housing).

**Main point 4 (points 1 & 2):**
205 The scene-setting needs to be done more carefully and the drafting of the hypotheses improved
   • More careful use of terminology needed; and some terms need defining.
      i. E.g. "flood protection" is more commonly used to mean property-level measures, but is not used in this way on p2. Perhaps "flood risk management" is a more appropriate term.
      ii. On p2 "private precautionary measures" is used before it has been defined.
210      iii. P2: the terms 'structural' and 'non-structural' need defining
   • 7.3: I think this should read "perceived cost of a protective measure…"

**Answer 4 (points 1 & 2):**
See Answer 3 (points 2-7): Important terms will be defined beforehand. As suggested, we will replace the term "flood
215 protection" with "flood risk management". Suggested changes in the text:

"The focus was shifted to a more integrated flood management, where also structural precaution measures (i.e. waterproof sealed cellars i.e. dry-proofing, wet-proofing, relocation of heating and electrical utilities) as well as non-structural flood protection measures (i.e. adapted interior fitting and flood-adapted use such as avoiding water-sensitive furniture in the cellar) became increasingly important (Kienzler et al., 2015; Thieken et al., 2016b; Laudan et al.,
220 2017)."

(…)

"Flood risk management in Germany has a long history with several regulations and ongoing programs."

225

**Main point 4 (points 3-5):**

• H1:

i. 'riverine' refers to the river bank; the term more commonly/accurately used for flooding from rivers is 'fluvial'

ii. some fluvial flooding is flashy, so the dichotomy presented is a false one. Are the authors talking about pluvial flashy floods only?

iii. what does it mean to say that flash floods are 'more dynamic'? This needs explication.

iv. is it an overgeneralisation to say that flashy floods are 'a bigger threat to life'? Where is the evidence for this assertion?

v. what is the provenance of this hypothesis: e.g. in theory or the literature?

• H2: like H1, this hypothesis requires anchoring in the literature. It also requires nuancing; after all, some negative psych impacts prompt greater likelihood of precaution.

• H3:

i. This is too vague. The reader needs to know which psych indicators are meant, and which psych characteristics

ii. Some explanation needed of the distinction between 'indicator' and 'characteristic' (and, later, 'manifestations').

**Answer 4 (points 3-5):**

These are important suggestions to improve the paper. We suggest the following changes in the text to clarify our motivation and sharpen the hypotheses:

"In contrast to river floods, flash floods emerge within (small) catchments where slopes are steep and defined, resulting in unpredictable flow dynamics that can be rough in terms of a high sediment transport, high flow velocities and forceful discharge (Borga et al., 2014). The forecast of such flood events is not yet reliable since they can develop with very short lead time. Apart from potentially high damage on buildings and infrastructure, flash floods can also cause serious injuries and fatalities (Gaume et al., 2009). Therefore it can be assumed that flash floods are perceived as a threat for personal health and property and induce negative psychological responses in flood experienced people."

(…)

"H1: Flash floods, in comparison to slowly emerging river floods, show a different psychological impact on affected people in which negative effects such as stress and feelings of being helpless are more pronounced, since flash floods are rough, emerge suddenly and therefore represent an unpredictable danger for health and property.

H2: Negative psychological impacts are connected to a lower probability for precaution because negative feelings might hamper the individual energy and self-confidence as well as the overall motivation to implement precaution measures.

H3: Psychological indicators such as the level of stress and coping appraisals are suitable for explaining precautionary behaviour because those psychological characteristics are distinctly connected to the protection motivation."

(…)

260    "Thus, groups of similar psychological characteristics (psychological indicators) are created first."

**Main point 4 (points 6 &7):**

265    • How is similarity defined when grouping 'similar psych manifestations' p4?

    • Lines 24-32 (p4):

        i. What is meant by 'an indicator'?

        ii. How can an indicator 'estimate' something? (Did the authors mean 'predict'?)

        iii. What is meant by a 'precaution level'?

270        iv. The authors need to justify their assumption that a better understanding of psych factors can inform 'targeted info campaigns'. It seems a little simplistic to think that information will make much difference. Plus, how would target groups be identified given that there's no easy way of identifying people with different psych characteristics.

**Answer 4 (points 6 & 7):**

275  The phrase "manifestations" will be replaced with "characteristics" since it fits better within this sentence. We will rewrite the respective test passages as follows:

    "•A good understanding of psychology and precaution motivation might result in a variable which indicates the probability for a good precaution and could be integrated into flood loss modelling and dynamic risk assessements as suggested by Aerts et al. (2018).

280

We will clarify our idea of targeted information campaigns. Indeed, the term is a little misleading and the targeting of specific societal groups seems challenging for the mentioned reasons. Here it is important that information campaigns also provide support and information for a broader audience and people who potentially need more help, support and specific info to match their needs. E.g. an information campaign with trained personnel might help to convince people with avoidant

285  tendencies. We will therefore rewrite the respective text passage:

    "•The outcome might be beneficial for  information campaigns that better support flood affected individuals in different flood prone regions. Various mental coping approaches could also be considered in such campaigns, since they may vary among different flood types and affected regions. The motivation to implement useful private flood precautionary measures could be strengthened according to the needs of individually affected people (e.g. Morss et al.,

290    2016)."

**Main points 5 & 6:**

5. The paper would benefit from some critique of Precaution Motivation Theory and a more sophisticated justification of its selection over other theories.

6. The paper needs to draw on literature from outside Germany. E.g. there is much on this topic from the UK. If the situation in Germany is so unique as to make other literature nonsalient (p3), this needs to be explained.

**Answer 5 & 6:**

Yes, such additions will improve the paper since the explanations and examples are currently too short. We therefore suggest following text changes/additions:

"Originally evolved in the health sector, the PMT gained attention in the domain of natural hazards over the years (Mulilis and Lippa, 1990; Grothmann and Reusswig, 2006; Bubeck et al., 2017). The model relies on two main cognitive processes - "threat appraisal" and "coping appraisal" – to describe the mental response to a specific threat. Threat appraisal is composed of the perceived consequences and probability of an event. Coping appraisal comprises the variables "self-efficacy" (perception of how well a person is able to carry out protection measures), "response efficacy" (how effective the measures are believed to be) and "response cost" (the perceived cost in terms of money and effort) (Rogers, 1975; Bubeck et al., 2012).

Main findings suggest that psychological factors – not only in terms of risk perception, but also avoidance and wishful thinking – can influence protective responses (Grothmann and Reusswig, 2006; Bubeck et al., 2012). Overall the PMT results in reliable estimations of protective behaviour, while particularly coping appraisal has been evaluated as a good predictor (Floyd et al., 2000; Milne et al., 2000; van Valkengoed et al., 2019). It has further been shown that the motivation to protect oneself from flooding cannot be solely explained by risk information, risk perceptions and socioeconomic factors such as income and homeownership (e.g. Baan and Klijn, 2004; Bubeck et al., 2012; Morss et al., 2016). Supportive evidence is given by Hopkins and Warburton (2015), who revealed that flash flood experience among UK citizens does not necessarily lead to higher risk perceptions. Yet, Harries (2012) shows that protective behaviour of flood affected UK citizens is significantly associated with the perceived probability to be flooded again while potential effects of protective behaviour such as feelings of safety; anxiety, and the fear of uninsurable impacts are influenced by flood experience. Having analysed flood affected households in Germany and France, Bubeck et al. (2018) identified good social norms and networks as an important factor for better coping abilities after river floods. Especially the trust in its own abilities and the belief in a good measure effectiveness increase with the number of neighbours, who already implemented flood protection measures. Eventually, these results suggest that among influencing factors on protective behaviour, psychological characteristics might play a significant role."

**Main points 7 (points 1-4):**

Findings

    • I do not believe it's appropriate to report as 'findings' correlations that are nonsignificant e.g. 11.4.

• Interpretations of statistical findings should be less speculative i.e. justified from theory/the literature. Presently, much of the interpretation appears no more than supposition. E.g. p11

• Interpretations of statistical findings are best reported separately from the findings themselves.

• Explanation required of some key terms: 'flood adapted use', 'better interior fitting' and 'damage ratio of buildings'
    (p8); 'coherence' (p9)

**Answer 7 (points 1-4):**

Thanks for the hints. We will rewrite the respective text passages and explain key terms. Suggested changes and additions in the text:

    "A similar  outcome is indicated when comparing the difference between strong flash floods and river floods, yet the results are not significant."

(…)

"The focus was shifted to a more integrated flood management, where also structural precaution measures (i.e. waterproof sealed cellars, relocation of heating and electrical utilities) as well as non-structural flood protection
    measures (i.e. adapted interior fitting and flood adapted use such as avoiding water-sensitive furniture in the cellar) became increasingly important (Kienzler et al., 2015; Thieken et al., 2016b; Laudan et al., 2017)"

(…)

"It has been shown that private precaution measures can significantly reduce the mean damage ratio (i.e. the financial flood damage in relation to the total building/content asset value) to households and household contents up to 53 per
    cent and thus play a significant role in comprehensive flood management (Kreibich et al., 2005; Thieken et al., 2008; Merz et al., 2010)."

(…)

"Yet, these results could be explained by the fact that people who were affected by strong flash floods believe similar events to be very unlikely to happen again in near future, resulting in lower feelings of threat. Although Hopkins and
    Warburton (2015) showed that flash flood experience does not necessarily lead to higher risk perceptions, it is unknown, to which degree lower feelings of threat are caused by a lower flash flood experience itself. Since almost all surveyed households experienced a strong flash flood for the first time (82%), they may not believe to be affected

again. However, an analysis of threat appraisal with corrected data in terms of flood experience (all households that experienced a flood for the first time) reveals a similar picture, i.e. threat appraisal is significantly lower for people who were affected by a strong flash flood in comparison to people who were affected by weak flash floods and river floods (see appendix, Figure D). This again supports the findings of Hopkins and Warburton (2015)."

Since the results and interpretation are closely connected within this study, we prefer to keep the section of combined results and discussion.

**Main points 7 (points 5-8):**
• Figure 2 appears to add little and is hard to understand. I suggest that it be removed or more carefully explained.
• Fig 3 – the x-axis requires more explanation
• 'JSD' requires spelling out p12 etc
• Discussion of the hypotheses:
  o H1: it needs to be made clearer how the suggested 'focus on threat perception' is justified by the findings. At present, the logical argument is weak/hazy.
  o H2: the text is v hard to understand.

**Answer 7 (points 5-8):**
We will explain all figures in greater detail and also improve the explanation in the paper. We will further define the acronyms and describe the methods in a clearer way beforehand. Suggested changes in the text and within the figure descriptions:

"The distributions of threat appraisal, coping appraisal, burden and evasion were further analysed using the Dunn's Test, which is based on the non-parametric Kruskal-Wallis rank sum test results. These tests are suitable for assessing the differences among the distributions of ordinal-scaled data, which does not fulfil assumptions of normality and equality of variance. Here, the Kruskal-Wallis rank sum test is preliminary to the Dunn's Test and calculates discrepancies among the rank sums of all values within the compared indicators. The derived Kruskal-Wallis statistic is then compared to the expected average difference among the sum of ranks via Dunn's Test. The outcome represents a measure for the disparity and shift of compared distributions. This approach reveals significant differences in psychological impacts which were predominantly caused by weak flash floods, strong flash floods and river floods."
(…)

390    Figure 2 description: " Figure 2: Relative distributions of the combined psychological indicators for each flood type and
       Dunn's Test results. The results of the Dunn's Test reveal the direction shift of each distribution compared to the other
       distributions (negative means a shift towards lower values, positive a shift towards higher values), by also indicating
       the strength and significance of the shift (Z-statistic and p-value)."

       (…)

395    Figure 3 description: "Figure 3: Relative distribution of the already implemented precaution indicator (left) and the
       planned precaution indicator (right) for weak flash floods (n=293), strong flash floods (n=116) and river floods
       (n=1366). The X axis represents the implementation of, or the intention to implement effective precaution measures.
       The higher the value, the more effective measures have been implemented, or will be implemented in near future. The
       indicator was based on results from Kreibich et al., (2005) and Thieken et al., (2005)."

400

Additionally, we will elaborate on the justifications regarding H1 and rewrite certain text passages regarding H2. Suggested
changes in the text:

       "Affected people perceive a strong flash flood event as less likely than people who have been repeatedly affected by
       river floods. Thus, future disaster risk management in Germany may also take into account that individual threat
405    perceptions of affected residents may differ from evidence-based hazard estimations, potentially leading to higher
       damage. Therefore, information campaigns in flash flood prone regions should be promoted, especially if various
       studies suggest an increase in severe flash flood events due to climate change and a change in weather patterns (e.g.
       Murawski et al., 2015)."

       (…)

410    "First, the assessment methods of psychological items as well as the items themselves do not follow established
       psychological assessment routines or surveys, what potentially decreases data consistency and accuracy. Second, subtle
       effects on precautionary behaviour that are caused by psychological aspects may be covered by incidental effects, due
       to the small sample sizes. This is particularly true for strong flash floods, leading to high uncertainties."

415

**Main point 8:**
       • The aim seems incorrectly described (16.11). Wasn't it the connection to precautionary behaviour that was explored,
       not that to motivation? (See H3)
420    • It would help if the key findings were foregrounded so that they were easier to pick out.
       • The authors should avoid making assumptions of causal direction (17.5). It's possible that preparedness influences
       frequency of remembering, rather than visa-versa (see Harries, 2008: "Feeling secure or being secure").

**Answer 8:**

425     Thanks for the suggestions and hints. We will adapt misleading descriptions. Further, the main results will be highlighted. We will alleviate the description of the overall result regarding the causal direction that was assumed. Suggested changes in the text:

"Further, the usefulness of psychological indicators and individual psychological variables to predict precaution behaviour was evaluated."

430     (…)

"Overall it is indicated that, in particular, the frequency of remembering an event is positively connected to preparedness intentions."

---

## Author Comment (AC2) · 3 Aug 2019

**Response to Anonymous Referee #2 RC2: nhess-2018-407-RC1, 2019**

Author: Jonas Laudan[1]
Co-authors: Gert Zöller[2], Annegret H. Thieken[1]

[1]University of Potsdam, Institute of Environmental Science and Geography, Karl-Liebknecht-Strasse 24-25, 14476 Potsdam, Germany
[2]University of Potsdam, Institute of Mathematics, Karl-Liebknecht-Strasse 24–25, 14476 Potsdam, Germany

We thank the reviewer for the constructive comments. We will extend our descriptions analysis and discussion as suggested by the reviewer to improve the paper quality.

**Comments of the reviewer**

**Reviewer quote, paragraph 1:**

A general limitation of the study seems to be that the people who experienced river floods experienced them multiple times in the last 10 years. While the people who experienced flash floods seem to have fewer past experiences. Is there a 20 possibility that this frequency of past experiences may be a stronger signal than the flood type? Is there a way to correct your data for the number of flood experiences people have had?

**Answer to paragraph 1:**

We agree that this is a general limitation of the study since the previous experience of flash floods is very low among the 25 surveyed residents. Yet, the residents who have been affected by river floods experienced several major flood events in recent years (2002, 2006/11, 2013) and therefore show a higher experience in total. However, not all regions that were hit by the 2013 flood, had been affected in 2002 or later, this holds particularly for Thuringia, Lower Saxony and Baden-Wurttemberg. The number of surveyed residents from these regions was, however, lower than aimed at. The correction of flood experience will decrease the flash flood data to a great extent, increasing analysis uncertainty. However, we will 30 analyse the indicator "threat appraisal" with regard to corrected data (all households that experienced a flood for the first time), discuss the results and put a figure (figure D) in the appendix.

**Reviewer quote, paragraph 2, 6, 7:**

Specific comments: Page 4, line 8-20, these are some very technical sentences, could you explain your approach in a more intuitive way and introduce the technical methods later. Currently this is difficult to read without prior knowledge about the statistical methods that are applied.

2.5 Explain why you use Bayesian statistics, you now just jump into the explanation without first motivating the choice.

2.5 Why did you choose to use Bayesian statistics if no prior is available? What is the advantage of using Bayesian statistics?

**Answer to paragraph 2, 6, 7:**

Thank you for this suggestion. We will rewrite sentences that are too technical and give explanations of the statistical methods beforehand. We will further elaborate the choice for Bayesian statistics. We decided to include Bayesian statistics without specific prior information since any valuable results from other studies could theoretically be integrated into our analysis in future. Suggested changes in the text:

"Secondly, the differences in the indicator distributions, i.e. shifts to lower or higher indicator ratings, are assessed for each flood type. To answer the second and third hypotheses, a "planned precaution" indicator is created first. In a next step, the Bayesian approach and negative binomial regressions are applied and resulting probability distributions of conditional variable dependences as well as regression coefficients are evaluated. The Bayesian approach has been frequently used in psychology (e.g. Wetzels et al., 2011) and other disciplines. It assesses the data uncertainty which is particularly helpful among studies that rely on relatively small data sets, while prior information independent of the data can be included (Van de Schoot et al., 2015). Since this study relies on small data sets, using the Bayesian approach as a supportive analysis helps to interpret main results. By revealing data and model uncertainties, the reliability of future prediction models that are based on these data sets can be evaluated in advance. Accordingly, this study considers Bayesian inference as a method to assess variable relations, that are based on conditional probabilities and related uncertainties. Preliminary assumptions such as e.g. linear variable coherences are therefore not required.  Bayesian statistics were also chosen due to the fact that the method enables prior knowledge to be taken into account, for example in following studies that use similar Bayesian approaches."

**Reviewer quote, paragraph 3:**

Page 4 line 24-30: Could you sketch in a bit more detail how you see this being used in the future. We don't know these psychological indicators for everyone when we make a damage model. It might even be easier to ask directly about precautionary measures than to assess their psychology. Using social media information as proxy might be a solution but I like to see these arguments made a bit more thoughtful and if that's the way to apply it I like to see that back in the discussion and maybe a recommendation to study how social media clues can be linked to the indicators used in this paper.

You mention several times the "protection motivation theory", please give a brief explanation of this, you can't assume all your readers know about this.

**Answer to paragraph 3:**

Thank you for the comment, we will elaborate on the topic of alternative data sources as well as new approaches to gather valuable data in the discussion. We will further give a brief outline of the Protection Motivation Theory beforehand. Suggested changes in the text:

"In this context, the protection motivation theory (PMT) (Rogers, 1975) has been frequently used as a psychological model to explain the risk-reducing/protective behaviour of affected individuals by analysing the influencing factors on coping strategies and potential positive or negative responses. Originally evolved in the health sector, the PMT gained attention in the domain of natural hazards over the years (Mulilis and Lippa, 1990; Grothmann and Reusswig, 2006; Bubeck et al., 2017). The model relies on two main cognitive processes - "threat appraisal" and "coping appraisal" – to describe the mental response to a specific threat. Threat appraisal is composed of the perceived consequences and probability of an event. Coping appraisal comprises the variables "self-efficacy" (perception of how well a person is able to carry out protection measures), "response efficacy" (how effective the measures are believed to be) and

"response cost" (the perceived cost in terms of money and effort) (Rogers, 1975; Bubeck et al., 2012)."

(…)

"• A better understanding of this connection might help to improve future vulnerability and risk estimations and may facilitate the use of alternative data sources to estimate the state of individual precaution. For example, data from online surveys, social media and communication platforms offers a lot of potential to assess individual mental coping strategies such as evasive behaviour or active remembering after severe events. With the help of advanced intelligent learning algorithms (e.g. random forests, neural networks and deep learning), psychological profiles could thus be created. Those might be used to develop sophisticated models and predict the state of precaution in areas which have not been flooded recently, all based on data given voluntarily by residents. Surveys that capture the state of precaution are still an alternative option."

(…)

"An issue of telephone surveys is that the data is becoming biased towards older participants when based on landlines (Greenberg and Weiner 2014). Alternatively, by implementing and making use of online surveys, smartphone applications and contracts with companies, valuable data could be collected accounting for people from all age groups. For further use, algorithms such as Neural Networks or deep learning algorithms may be applied on this data to create or categorize psychological aspects such as the expected level of burden or evasion in case of an event. Those techniques might result in good predictions of psychological behaviour and the connected precaution motivation and can theoretically be transferred to other regions but yet imply certain challenges. Firstly, large amounts of consistent and high quality data have to be collected on condition that data security and personal rights are considered. Secondly, the interpretation of results in terms of causality and meaning is hampered due to the black box character of the analysis, even though potential results might show a certain robustness."

**Reviewer quote, paragraph 4 & 5:**

Page 7, line 5, please first explain what burden and evasion are before explaining the motivation behind it.

Give a proper explanation of Kruskal-Wallis rank sum test, Dunn's Test, the Jensen-Shannon divergence and regression tests directly after you first mention these methods. Maybe don't mention them too early in the text. Give both an intuitive and a brief mathematical explanation of the methods.

**Answer to paragraph 4 & 5:**

As you suggested, we will describe and explain the methods as well as key terms beforehand and in a clearer way. Suggested changes/additions in the text:

"Subsequently, the four main indicators are defined as "threat appraisal", "coping appraisal", "burden" and "evasion", which also show low intercorrelations and offer a certain comparability to other studies. The four indicators are thus defined and created as follows.

According to the PMT, threat appraisal consists of the perceived probability of being affected again by a flood event and the perceived impact of such a future event. Coping appraisal comprises self-efficacy, response efficacy and response cost which describes the self-rated ability to implement a protective measure, the perceived efficiency of a protective measure and the perceived cost of the protective measure, respectively (Grothmann and Reusswig, 2006;

Bubeck et al., 2012).

Burden describes a measure for the negative psychological load of the experience and consists of the single variables "often thinking of the event" and "stress still today". Evasion comprises the variables "avoidance" and "fatalism" and can be seen as a measure for the effort to get the experience out of one's mind for various reasons. Burden and evasion were developed by following the general procedure in psychology surveys to combine expressive psychological items (e.g. Ware and Sherbourne, 1992; Kroenke et al., 2001) and taking high correlations among psychological variables into account."

(…)

"The distributions of threat appraisal, coping appraisal, burden and evasion were further analysed using the Dunn's Test, which is based on the non-parametric Kruskal-Wallis rank sum test results. These tests are suitable for assessing the differences among the distributions of ordinal-scaled data, which does not fulfil assumptions of normality and equality of variance. Here, the Kruskal-Wallis rank sum test is preliminary to the Dunn's Test and calculates discrepancies among the rank sums of all values within the compared indicators. The derived Kruskal-Wallis statistic is then compared to the expected average difference among the sum of ranks via Dunn's Test. Similar to a power analysis, the effect size and significance are revealed for a given sample size. The outcome represents a measure for the disparity and shift of compared distributions. This approach reveals significant differences in psychological impacts which were predominantly caused by weak flash floods, strong flash floods and river floods."

Regarding the Jenson-Shannon divergence and the negative binomial regression, the respective parts in the text will be converted in a way that the explanation follows directly after mentioning the methods for the first time.

Figure 2: why is threat appraisal lowest for strong flashfloods? Does it make sense that if something extreme happened to you, you feel the probability that it will happen again to be lower? (your argument on page 11, line 10). Maybe threat appraisal is lower because they only experienced it once while the river floods and weak flash floods were experienced more frequently. If however you would go to another region where only one weak flash flood or river flood was experienced these results may look very different. You should probably discuss that limitation in the study.

**Answer to paragraph 8:**
We agree with the statement of the reviewer that this means a certain limitation of the study. Therefore, we will add another analysis in the appendix with the corrected data in terms of flood experience for the indicator "threat appraisal" (see also Answer to paragraph 1). The limitation will be discussed in a more elaborate way, yet we believe that our general statement in that case ('it has been such an extreme event that people perceive it as unlikely to happen again') holds true. Suggested changes in the text:

"Although Hopkins and Warburton (2015) showed that flash flood experience does not necessarily lead to higher risk perceptions, it is unknown, to which degree lower feelings of threat are caused by a lower flash flood experience itself. Since almost all surveyed households experienced a strong flash flood for the first time (82%), they may not believe to be affected again. However, an analysis of threat appraisal with corrected data in terms of flood experience (all households that experienced a flood for the first time) reveals a similar picture, i.e. threat appraisal is significantly lower for people who were affected by a strong flash flood in comparison to people who were affected by weak flash floods and river floods (see appendix, Figure D). This again supports the findings of Hopkins and Warburton (2015)."

Figure 4: This figure is not very intuitive can you explain a bit more what the reader sees here.

Figure 4: Why do you see the double peaks in the probability distributions?

**Answer to paragraph 9 & 10:**

Thank you for the hint. Firstly, we refer to the explanation graphic, Figure 1, where we will add more details to the description. We will also describe the method in a better way, changing the respective text passage underneath Figure 4. Suggested changes in the text:

"Figure 1: Example graphic explaining the creation of the weighted arithmetic mean posterior. The double peaks are a
result of the combination of all posteriors in one plot that are calculated for each variable combination. The posteriors are weighted according to the sum of occurrences within the dataset. In this case the weighted mean posterior means that, given the example dataset of 20 data points, it is most likely that a specific predictor variable rating occurs together with only one specific response variable rating to 80%."

(…)

"The weighted arithmetic means of all posterior distributions reveal in general a wide range of likely probabilities for the conditional dependence of variable ratings. In the case of weak flash floods for example, it is second most likely (second highest posterior peak) that a particular burden rating is always reported together with a specific rating of the planned precaution to 52 per cent (most likely to 9 per cent due to the highest posterior peak at this point). For coping appraisal, the most likely percentage would be 7 per cent. For threat appraisal and evasion, the most likely percentages
are 10 and 19 per cent, respectively (Figure 4, top left)."

Page 14, line 17: You say this is common practice in psychology, can you provide a reference for this?

**Answer to paragraph 9 & 10:**

A reference will be added. Suggested changes in the text:

"When comparing the analysis of the psychological indicators and the single variables, it can be summarised that a
combination of items, as it is practised by e.g. Ware and Sherbourne (1992) and Bei et al. (2013), does not lead to more
consistent and meaningful results in this case which is mainly reflected by similar JSDs."

**Reviewer quote, paragraph 9 & 10:**

Page 15, H2: I think the findings make much more sense than the hypothesis.

**Answer to paragraph 9 & 10:**

We think that the Hypothesis is justified since personal experience and conversations with flood affected residents indicated
a high level of burden after a severe flash flood event, which could also lead to negative responses and low motivation deal
with any aspects and implications of the flood event again. Still, we believe that these are interesting negative results which
support other studies such as Bei et al. (2013), who reported that affected people with worse mental and physical health show
a higher willingness for coping strategies.

---

## Referee Report (RR1)

**Flash floods versus river floods  a comparison of psychological impacts and implications for precautionary behaviour**

Reviewer comment on nhess-2018-407

December 19, 2019

The authors have put quite some effort into incorporating recommendations made on previous versions of this manuscript, which is commendable.

As pointed out by Referee #1 in the previous iteration, I am still struggling with some parts of the text, which remain hard to read and understand – specifically Section 2. I would strongly advise to revise this section again with a focus on accessibility.

Some passages seem to be framed in a strange way throughout the text. For instance, Section 2.4 starts with 'To apply the Bayesian statistics and regression models, an indicator (. . . ) had to be derived'. This implies that using a certain Bayesian approach is the main goal rather than actually answering a research question.

In the following, I have focused specifically on Section 2 (Data and methods):

- Section 2.3 (p9 l26ff): The authors might want to add that 'correlation' refers to rank correlation.

- Section 2.3 (p9 l27): The RStudio Version is not that relevant, since this is merely an IDE. If versions are reported, please report the R version and package versions instead.

- Section 2.3 (p9 l29): Albeit this is subject to subjectivity, I am not sure if I would call 0.54 to be a 'strong' correlation.

- Section 2.3 (p9 l32): Please note that calculating statistical power based on the observed effect size after the study has been carried out is fundamentally flawed. After the study, reporting confidence intervals for effects (ideally) or p-values is the proper way to present results.

- Section 2.3 (p10 l10): I am not sure if 'preliminary' is the proper word to use in this context.

- Section 2.4: I find this section very difficult to follow. There are lots of complex multi-clause sentences which left me quite confused. Upon reading the section multiple times, I think I finally know what the authors actually did, but this should be clear to the reader when reading this section the first time. I guess that some of the confusion is caused by the terminology - the authors mix the terms 'indicator', 'measure' and 'score' quite a bit throughout this section. We want to derive an indicator for planned precaution, which is derived from flash flood and river flood data sets (p10 l18). This indicator is based on existing studies. Then we suddenly have two indicators in this paragraph (planned precaution and already implemented precaution, p10 l23), consisting of measures (which measures?), which are weighted (how?) according to their damage potential. In the next sentence (p10 l25) 'it' (what is 'it' exactly? There are two indicators in the preceding sentence), resembles a score. This score of weighted measures (p10 l28) is summed up and related to measures (p10 l30 - shouldn't the score compared to the score and not the measures?) implemented before the event as well as missing answers (How can something be compared to missing answers?). Please streamline this section and try to clarify the procedure.

- Section 2.4: Also, I am under the impression that quite some information might be lost in constructing the indicator by first limiting the count to 8 and then reclassifying the resulting score (on a sidenote: I assume that the reclassification is based on equal interval sizes, but this is not described in the text).

- Section 2.5: Elements of Equation (1) are not explained in the text. Also, please note that likelihood is not called $L$ in the equation, as mentioned in the text below the formula.

- Section 2.6: Please refer to Shannon-Entropy with respect to Equation (3) before defining it in Equation (4). It cannot be assumed that all readers are familiar with this concept (p12 l29, 'Where'). In addition, I think that the formula for Shannon Entropy is not clear. What is $i$ in this context? The base of log is not clear either. I think it should rather read something like $\mathrm{H}(X) = -\sum_{i=1}^{n} \mathrm{P}(x_i) \log_b \mathrm{P}(x_i)$ with $X$ being a discrete random variable with possible values $\{x_1, \ldots, x_n\}$ and probability mass function $\mathrm{P}(X)$.

---

## Author Response (AR2)

**Response to Referee #1, Tim Harries, 2019**

Author: Jonas Laudan[1]
Co-authors: Gert Zöller[2], Annegret H. Thieken[1]

[1]University of Potsdam, Institute of Environmental Science and Geography, Karl-Liebknecht-Strasse 24-25, 14476 Potsdam, Germany
[2]University of Potsdam, Institute of Mathematics, Karl-Liebknecht-Strasse 24–25, 14476 Potsdam, Germany

We thank the reviewer again for his review and comments how to improve the manuscript.

**Comments of the reviewer**

My thanks to the authors for their efforts to take on board the recommendations made by myself and the other reviewer. In particular, the description of the methods has improved considerably and some of the text has become easier to read. Unfortunately, I still have a number of serious reservations, especially regarding the scientific significance of the paper, the attribution of participants to groups (weak, medium and strong) and the clarity of the writing.

**Main point 1:**

1. Hypotheses. In order to contribute to the extant literature and have scientific significance, the study needed to have drawn on that literature for its hypotheses. I do not feel this has been done/done sufficiently.

• H1: I am not aware of any empirical evidence to suggest H1. Neither I am convinced by the common-sense justification provided by the authors. Hence, I do not see the value of expending so much effort (and reader time) on conducting a test that fails to support it.

• H2 is more interesting but poorly described and, like other Hs, not grounded in the literature. Where is the research evidence to suggest that negative psych impacts will reduce the motivation to implement precaution? I would have liked to see a presentation of the literature on this issue. On p17 the authors themselves cast doubt on the scientific value of the way they teted this hypothesis - when they assert that the sample size was insufficient for H2 and that established processes were not followed.

• H3 remains poorly described. The reader needs to know which psych indicators are being tested rather than just being given examples. It is not clear what is meant by 'are suitable' and this cannot be tested with statistics. What does 'distinctly connected' mean?

**Answer 1:**

• To our knowledge, Bei et al. (2013), Gaume et al. (2009) as well as Mason et al. (2010) already provide evidence that severe flash floods describe a danger not only to property but also to physical and mental health and therefore justify hypothesis 1. Additional empirical evidence is now given in the paper for a better grounding of all hypotheses.

Paper changes:

(Page 14, line 30) This effect is further described and underpinned by Hudson et al., (2019), who found out that flood experience is connected to a loss in subjective well-being among flood affected residents in Vietnam, while females tend to recover slower than males. This was also found by Bubeck and Thieken (2018) for Germany. Additional evidence for negative mental health effects after floods is given by Wagner (2007), who suggests that models of anxiety and coping can be related to fears of different hazard types. Those models describe reactions and coping strategies of people whoichwho are guided by vigilance (i.e. actively searching for threat-related information) and avoidance (i.e. denial & distraction). Moreover, a comprehensive review of Fernandez et al. (2015) on flood related mental health issues as well Foudi et al. (2017) strongly support the assumption that, in case of flood exposure, especially water depth and high flow velocities have a negative impact on mental health in terms of increased levels of PTSD, anxiety, as well as depression. This is also supported by Lamond et al. (2015) who suggest that psychological symptoms such as stress and anxiety remain as a result of severe flooding and flood damage. Further they reveal that mental health issues are related to post-flood mitigation actions, where especially relocating seems to be a suitable measure.

• By mentioning problems and methodological challenges, we do not intend to disqualify our work, but suggest valuable improvements and giving thoughts to an improved design of future studies.

• For clarification we rewrote H3 in the following way:

(Page 15, line 29) Psychological indicators such as e.g. the feelings of stress and burden that people still perceived at the time of the interview or self-reported coping abilities can be used as a proxy to explain precautionary behaviour because such mental feelings and attitudes are connected to the motivation and intention to protect oneself in future as highlighted e.g. by the Protection-Motvation-Theory and others.

**Main point 2:**

There is no shame is producing a negative result, yet the authors seem to try to hide this. E.g. if you have 'low explanatory power' and 'non-significant results' this suggests that the psych indicators have no usefulness rather than 'limited usefulness'.

5 **Answer 2:**

We agree that there is no shame in producing negative results and support the approach to avert post-hoc hypothesis adjustments in general. We rewrote the mentioned paragraphs in a way that any concerns about alleged hiding intentions may be eliminated.

Paper changes:

10      (Page 11, line 15, Abstract) Aaccording to the used data, however, predictions of the individual precaution motivation should not be based on the derived psychological indicators, i.e. "coping appraisal", "threat appraisal", "burden", and "evasion", since their explanatory power was generally low and results are, for the most part, non-significant. Only burden reveals a significant positive relation to planned precaution regarding weak flash floods. A remarkable difference to weak flash floods and river floods can be seen with regard to strong flash floods, where the
15      perceived threat is significantly lower although feelings of burden and lower coping appraisals are pronounced.

**Main point 3:**

20 The non-significance (rather than 'low significance') of the relationship between avoidance and fatalism amongst those experiencing strong flash floods seems to call into question the validity of combining avoidance and fatalism when looking at strong flash floods. Yes, this 'may' be due to the sample size, but where is the statistical evidence to support this suggestion?

25 **Answer 3:**

The variable combinations and resulting indicators have been chosen to remain the same among different flood types to ensure a certain comparability. Therefore we also combined avoidance and fatalism among those who experienced strong flash floods. To gain insights into single variable connections we performed a correlation analysis of the planned precaution and all separate psychological variables. Here it is revealed how they perform as predictors on their own.

30 In fact, we accidently reported a wrong p-value regarding the correlation between avoidance and fatalism of strong flash floods. The new corrected p-value is 0.07 and therefore significant at a <0.1 level.

Still, for a better clarity we will include a power analysis to test the correlations.

Paper changes:

(Page 20, line 34) and a power of 1.0 in all cases. Further, avoidance and fatalistic thoughts reveal a correlation of 0.23 (complete cases n=275, $p<0.05$, power=0.97) for weak flash floods, 0.29 (complete cases n=113, p=0.34 p=0.07, power=0.88) for strong flash floods and 0.18 (complete cases n=1242, $p<0.05$, power=1.0) for river floods.  Therefore, we combined avoidance and fatalistic thoughts as two different strategies of mal-adaptive behaviour

**Main point 4:**

The categorisation used within the study aim and the paper title remains problematic. Flash floods and river floods are not mutually exclusive categories. Rivers can themselves produce flash floods (e.g. when their catchment has a low absorptive capacity). I may be wrong, but I think that it would be more accurate to distinguish "flash" floods from "slower onset" floods and that the source of the flooding (pluvial or fluvial) is irrelevant to their question.

**Answer 4:**

It is true that the boundaries of flash floods and river floods can be rather fluid. However it is also true that orography describes an important aspect to consider in potential flash-flood-prone regions. Our flood data is based on two different datasets after two different events. One survey has been conducted after a strong river flood event in 2013 with slow onset times, the other contains only persons affected by heavy rainfall and following floods with a short lead time, a more or less forceful runoff (pluvial floods/urban flooding) and high flow velocities. According to this, we distinguished the flood types as described in the paper.

**Main point 5:**

Large parts of the text remain hard to read and, in many instances, hard to understand even after repeated reading. E.g. lines 25-32 on p4; 29-31, p7. It is not uncommon for the former to indicate that authors themselves do not fully understand what it is that they are trying to express, so the lack of clarity makes me particularly nervous. There are two main issues here: logical clarity (i.e. in the construction of sentences and paragraphs) and use of English.

Terminology. "Flood dynamics" needs to be defined before it is used.

**Answer 5:**

Additional proofreading by a native English speaker was performed before the resubmission of the revised paper. Changes are included in the revised version but not detailed in this response letter. We hope that this process has improved the readability of the paper.

**Main point 6:**

Methods

10 • In an ideal world, all other factors should be kept constant in a test of the impacts of type of flood on psych. The confession that the two regions "differ almost completely" is therefore a peculiar one; it seems to invalidate the whole exercise.

• The two surveys are described as "very similar". I would want some reassurance that none of the differences impact on the validity of the analysis.

• Was "information gathering" aggregated with other precautionary measures in the outcome variable used? Why? My
15 reading of the overall text suggests that the intended/actual precautions variables refer to tangible measures and not to information seeking. This should perhaps be made clearer.

• The attribution of weak, strong and medium to the flash floods is key to the analysis, so needs careful explanation. I find this a particularly vulnerable aspect of the research design and need reassurance and greater clarity. I suggest more detail on this aspect.

20 • The authors should make it explicit that weak, strong and medium relate to impacts on the areas and do not necessarily reflect levels of physical impact on particular homes/residents.

• Given that online literature and the press usually report only the most dramatic floods, I wonder how the authors were able to identity "weak" floods. It would not be safe to assume that any flood not mentioned in the press was "weak" as it is unlikely that all strong/medium floods were actually reported on.

**Answer 6:**

• It is theoretically correct, but practically questionable to assume that regions as well as the flood types should not differ, since the characteristics of flood types we considered are -besides rainfall intensity- also influenced by orography and catchment sizes. Despite the fact that river floods and flash floods are not mutually exclusive, they rarely affect the same
30 region and therefore different regions and residents are affected by different floods within a reasonable time span. As stated by Wagner (2007), we agree that the region, or more precise the local condition, is a relevant factor of risk perception. However after being affected by a flood, behavioural aspects and individual mental coping are influenced much more by the recent event itself and depend on personal character traits, rather than the particular region. Of course, longitudinal studies

with the same persons being repeatedly affected by different flood types would be desirable but yet, data scarcity is an issue and panel data are challenging to collect as recently outlined by Hudson et al. (2019).

• Yes, we are confident that none of the differences impact on the validity of the analysis and we clarified this within the paper.

• Information gathering, too, factors into the 'planned precaution indicator' and it is true that not only tangible measures have been considered. We clarified this that in the paper, as well.

• We further elaborated on the fact that the flash flood strength is considered for a larger area, in most cases a whole municipality, and not at particular houses.

• To not rely solely on press articles we compared the reported event with rainfall data from DWD (Deutscher Wetterdienst) - as described in the paper - for a better judgement of the local flood severity.

Paper changes:

(Page 17, line 27) The outline of both surveys is identical regarding all questions that were chosen for this study. In general, the questionnaires (…)

(Page 19, line 5) The flash flood strength was assessed on the municipality scale. It can be assumed that impressions and effects of the flash flood severity are not particularly dependent on the intensity at the individual house but are rather influenced by the overall appearance and effects of the flood within a village/town, which also includes impacts on neighbours, friends and infrastructure. It makes sense that mental coping, especially after strong flash floods, is not solely influenced by the individually experienced damage but dependent on experiences gathered in the local neighborhood.

Moreover, not only the impact, but also the potential to be harmed outdoor in case of sudden and strong flow forces may influence the mental coping in regions which can experience strong flash floods. In this context, Morrs et al. (2016) showed that people who perceive flash floods as a risk to their life tend to protect themselves if they receive a flash flood warning. Therefore it can be assumed that the mental impacts after a severe event are differing with regard to the severity within an affected area (eg. Bei et al., 2013).

(Page 19, line 25) as well as associated rainfall in the area at the particular time based on data from DWD, (…)

(Page 21, line 28) It resembles a score of precaution in which information gathering, non-structural precaution structural precaution and preparation are included and weighted according to their effectiveness (see section 2.1 for the private precaution measures).

Given that the paper places such emphasis on the role of denial, it seems strange that the authors do not utilise 'denial' to
explain lower feelings of threat amongst those experiencing strong floods and that they treat respondents' answers to survey
questions as beliefs – p12. On p14 (line 21), the authors make the cardinal error of reading causal direction into correlation.

**Answer 7:**

"Denial" is a variable that we cannot derive from our dataset with absolute certainty and we also did not record it as a
standalone item. For this reason it is not possible to utilise it in such ways. However, the role of denial should definitely be
considered in follow-up studies.

Concerning Threat appraisal, we mention in the text that the given ratings are based on perceptions. More specifically it is
asked, how people perceive the probability to be impacted again by a flood and if they think that the event will be severe
again. See text in paper:

> (Page 13, line 32) PMT relies on two main cognitive processes - "threat appraisal" and "coping appraisal" - to
> describe the mental response to a specific threat. Threat appraisal is composed of the perceived consequences and
> probability of a specific threat. Coping appraisal comprises the variables "self-efficacy" (perception of how well a
> person is able to carry out protection measures), "response efficacy" (how effective the measures are believed to be)
> and "response cost" (the perceived cost in terms of money and effort) (Rogers, 1975; Bubeck et al., 2012).

> (Page 28, line 26) Here, sperceived feelings of burden  are positively related to a higher precaution
> motivation.

**Main points 8:**

To make section 3.3 more accessible to readers not familiar with Bayesian analysis, the authors might want to explain to
readers the meaning and significance of the term 'posterior distribution'.

**Answer 8:**

We will elaborate on that to highlight the significance of the Bayesian analysis.
Paper changes:

> (Page 23, line 4) This means that a given variable value (e.g. score 3 out of 6 possible scores among planned
> precaution) occurs with a particular value of another variable (e.g. score 6 out of 6 possible scores among
> avoidance) to most likely e.g. 45 per cent (peak of parameter p).

(Page 23, line 19) In detail, this means that the combined posterior distribution shows the likeliness of the likeliness of all mutually occurring variable scores (or values) in a single graph. Here the distribution shape of parameter p (i.e. its highest peak) resembles the most likely probability of mutual occurrence, in the dataset at hand.

**Main points 9:**

Conclusion section

• "Individual threat perceptions differ from evidence based hazard estimations" p16. This seems obvious and is certainly not a new finding.

• The recommendation of 'information campaigns' seems naive given the sophistication of the psychology in this paper. Given the complex emotion regulation that characterises the response to flooding and flood risk (e.g. denial), it is hard to imagine that it will be enough to simply tell people that they might be flooded again. See my 2008 and 2018 papers in Health, Risk & Society and International Small Business Journal for some additional insights.

• I'm afraid I find little in the Conclusion section that adds to existing knowledge of this topic.

**Answer 9:**

Of course, only information does not necessarily lead to an adaptive response of flood affected individuals. This is already included in psychological models. Therefore we refer to information campaigns in a more comprehensive way.

Paper changes:

(Page 16, line 27) The outcome might be beneficial for information campaigns that better support flood affected individuals in different flood prone regions. Various mental coping approaches could be considered in such campaigns, since they may vary among different flood types and affected regions. The motivation to implement suitable private flood precautionary measures could be strengthened according to the needs of individually affected people (e.g. Morss et al., 2016) This could be achieved by strenghtening the beliefes in precautionary measures, informing about the risk and offering mental support. For heavy rainfalls that lead to pluvial floods as well as for river floods, examples on precaution from the neighborhood could be communicated in combination with risk maps for specific areas. Regarding strong flash floods it could be meaningful to include affected people in strategies that can be realised on municipality level (e.g. retention areas), highlighting the dangers of such events and informing about specific private precaution measures that could lower building damage.

**Response to Anonymous Referee #2 RC2, 2019**

Author: Jonas Laudan[1]
Co-authors: Gert Zöller[2], Annegret H. Thieken[1]

[1]University of Potsdam, Institute of Environmental Science and Geography, Karl-Liebknecht-Strasse 24-25, 14476 Potsdam, Germany
[2]University of Potsdam, Institute of Mathematics, Karl-Liebknecht-Strasse 24–25, 14476 Potsdam, Germany

We thank the reviewer again for the constructive comments that helped to further improve the paper.

**Comments of the reviewer**

In general I'm very happy with the revisions and think the paper is almost ready for publication. Just the following three points need some attention:

**Reviewer main point 1:**
In the introduction you mention "negative binomial regressions", maybe add a sentence explaining what that is.

**Answer to main point 1:**
Thank you for the overall positive response to our revisions.

A short explanation regarding the "negative binomial regression" is now provided in the revised version.

Paper changes:

> (Page 16, line 9) Negative binomial regression can be used to model ordinal count data where variance and mean are not equivalent.

**Reviewer main point 2:**
Throughout the paper the Jensen-Shannon divergence is applied. I think the explanation of this method
is still a bit weak, the connection to the Kullback-Leibler divergence that is not explained isn't very useful.
Also not all parameters in formulas 3 and 4 are defined.

**Answer to main point 2:**

Thank you for this hint, we carefully revised the whole methods section and hope that it is more accessible for readers now.

Paper changes:

(Page 24, line 2) P = posterior distribution, R = reference posterior distribution

(Page 24, line 9) The divergence represents the degree of mutual information between two or more variable distributions and the strength of their connection (or to which degree they are distinguishable). Consequently, the JSD was used to assess the similarity of each posterior distribution and its reference posterior distribution to reveal if they differ from each other. The JSD can take any value between 0 and 1. If the JSD of the reference posterior and the calculated posterior is 0, both underlying variables (e.g. the planned precaution indicator and burden) are independent from each other and do not show any relation apart from random effects. If the JSD is greater than 0 however, these variables show a certain information gain if one is explained by the other. If the JSD is 1, both underlying variables are identical.

**Reviewer main point 3:**

Figure D in the appendix, please provide a legend indicating what color is what flood type.

**Answer to main point 3:**

Thank you for this hint. A legend was added.

[revised manuscript text omitted]

30 weighted according to their effectiveness (see  section 2.1 for the private precaution measures).

For the planned precaution indicator, this score of measures which were directly implemented during the event (presumably in connection with emergency measures), or planned to be implemented  shortly (up to 6 months) after the flood event  is summed up and related to the measures implemented already before the event

Kommentiert [JOL10]: Changes, main point 3, RC1

Kommentiert [JOL11]: Changes, main point 6, RC1

[revised manuscript text omitted]

---

## Author Response (AR3)

**Response to anonymus Reviewer, comments from December 12th, 2019**

Author: Jonas Laudan[1]
Co-authors: Gert Zöller[2], Annegret H. Thieken[1]

[1]University of Potsdam, Institute of Environmental Science and Geography, Karl-Liebknecht-Strasse 24-25, 14476 Potsdam, Germany
[2]University of Potsdam, Institute of Mathematics, Karl-Liebknecht-Strasse 24–25, 14476 Potsdam, Germany

We thank the reviewer again for his review and comments how to improve the manuscript.

**Comments of the reviewer:**

The authors have put quite some effort into incorporating recommendations made on previous versions of this manuscript, which is commendable.

As pointed out by Referee #1 in the previous iteration, I am still struggling with some parts of the text, which remain hard to read and understand - specifically Section 2. I would strongly advise to revise this section again with a focus on accessibility.

Some passages seem to be framed in a strange way throughout the text. For instance, Section 2.4 starts with `To apply the Bayesian statistics and regression models, an indicator (...) had to be derived'. This implies that using a certain Bayesian approach is the main goal rather than actually answering a research question.

In the following, I have focused specifically on Section 2 (Data and methods):

**General answer:**

According to your review we checked the complete manuscript again to improve readability and understandability. Large sections have been rewritten and relocated and we hope that especially the clarity of Section 2 has improved.

**Point 1:**

Section 2.3 (p9 l26ff): The authors might want to add that `correlation' refers to rank correlation.

**Answer 1:**

We agree and are now more specific when explaining the correlation method in Section 2.3.

**Point 2:**

Section 2.3 (p9 l27): The RStudio Version is not that relevant, since this is merely an IDE. If versions are reported, please report the R version and package versions instead.

**Answer 2:**

Instead of the RStudio version, the R version and package versions are now reported in the paper.

**Point 3:**

Section 2.3 (p9 l29): Albeit this is subject to subjectivity, I am not sure if I would call 0.54 to be a `strong' correlation.

**Answer 3:**

We adjusted this interpretation in the text.

**Point 4:**

Section 2.3 (p9 l32): Please note that calculating statistical power based on the observed effect size after the study has been carried out is fundamentally flawed. After the study, reporting confidence intervals for effects (ideally) or p-values is the proper way to present results.

**Answer 4:**

Thank you for this suggestion. In the revised version, , we reported confidence intervals for the correlation results to facilitate the analysis interpretation.

**Point 5:**

Section 2.3 (p10 l10): I am not sure if `preliminary' is the proper word to use in this context.

**Answer 5:**

5   According to your hint, we rewrote and shortened the sentence.

**Point 6:**

10   Section 2.4: I find this section very difficult to follow. There are lots of complex multi-clause sentences which left me quite confused. Upon reading the section multiple times, I think I finally know what the authors actually did, but this should be clear to the reader when reading this section the first time. I guess that some of the confusion is caused by the terminology - the authors mix the terms `indicator', `measure' and `score' quite a bit throughout this section. We want to derive an indicator for planned precaution, which is derived from flash flood and river flood data sets (p10 l18). This indicator is based on existing
15   studies. Then we suddenly have two indicators in this paragraph (planned precaution and already implemented precaution, p10 l23), consisting of measures (which measures?), which are weighted (how?) according to their damage potential. In the next sentence (p10 l25) `it` (what is `it` exactly? There are two indicators in the preceding sentence), resembles a score. This score of weighted measures (p10 l28) is summed up and related to measures (p10 l30 - shouldn't the score compared to the score and not the measures?) implemented before the event as well as missing answers (How can something be compared to missing
20   answers?). Please streamline this section and try to clarify the procedure.

**Answer 6:**

According to your remarks, we have rewritten the whole Section 2.4 and used a more consistent terminology to increase clarity of the description. We explained the indicators in a better way and reported in detail on the method, how variables had been
25   combined and weighted. The seciton now reads as follows:

    **Indicator on precautionary behaviour**

[revised manuscript text omitted]

**Answer 7:**

Thank you for making that point. In general, the procedure of indicator creation (summing up implemented or planned measures) leads to much higher scores due to weighting (up to values of 48) which then have been reclassified (based on equal interval sizes) to range between 0 and 8. This classification was done to reduce the information to its most important aspects

10 and to enable correlation tests for ordinal data as well as the Bayesian approach. We rewrote the respective text passage for a better understanding of this issue (see also Answer 6).

15 **Point 8:**

Section 2.5: Elements of Equation (1) are not explained in the text. Also, please note that likelihood is not called L in the equation, as mentioned in the text below the formula.

**Answer 8:**

20 We rephrased the section according to your suggestions. It now reads as follows:

The statistical model takes prior knowledge into account ($P_0$(model parameter), also called prior) and assesses the likelihood to observe the data, if specific model parameters are given (P(data|model parameter)). This results in a probability density for the model parameters, conditioned on specific data (P(model parameter|data), also called

25 posterior) (Puga et al., 2015). The underlying principle is Bayes theorem, which given the above is show in equation (1):

$$P(model\ parameter|data) \sim P(data|model\ parameter) * P_0(model\ parameter) \qquad (1)$$

30 The likelihood P(data|model parameter) is based on the binomial distribution for each response variable (planned precaution) and predictor variable value.

**Point 9:**

Section 2.6: Please refer to Shannon-Entropy with respect to Equation (3) before defining it in Equation (4). It cannot be assumed that all readers are familiar with this concept (p12 l29, `Where'). In addition, I think that the formula for Shannon Entropy is not clear. What is i in this context? The base of log is not clear either. I think it should rather read something like

5   $H(X) = -\sum_{i=1}^{n} P(x_i) \log_b P(x_i)$ with X being a discrete random variable with possible values $\{x_1,...,x_n\}$ and probability mass function P(X).

**Answer 9:**

We rewrote the section according to your suggestions. The new version reads:

[revised manuscript text omitted]

For the planned precaution indicator, this score of measures which were directly implemented during the event (presumably in connection with emergency measures), or planned to be implemented shortly (up to 6 months) after the flood event is summed up and related to the measures implemented already before the event as well as missing answers (NA). 
[revised manuscript text omitted]

